# ANCHOR: Abductive Network Construction with Hierarchical Orchestration for Reliable Probability Inference in Large Language Models

**Wentao Qiu** [* 1]  **Guanran Luo** [* 1]  **Zhongquan Jian** [2]  **Jingqi Gao** [1]  **Meihong Wang** [1]  **Qingqiang Wu** [1 3 4 5]

## Abstract

A central challenge in large-scale decision-making under incomplete information is estimating reliable probabilities. Recent approaches use Large Language Models (LLMs) to generate explanatory factors and coarse-grained probability estimates, which are then refined by a Naïve Bayes model over factor combinations. However, sparse factor spaces often yield "unknown" predictions, while expanding factors increases noise and spurious correlations, weakening conditional independence and degrading reliability. To address these limitations, we propose ANCHOR, an aggregated Bayesian inference framework over a hierarchical factor space. It constructs dense factor hierarchies through iterative generation and clustering, maps contexts via hierarchical retrieval and refinement, and augments Naïve Bayes with a Causal Bayesian Network to model latent factor dependencies. Experiments show that ANCHOR markedly reduces "unknown" predictions and produces more reliable probability estimates than direct LLM baselines, achieving state-of-the-art performance while significantly reducing time and token overhead.

## 1. Introduction

Large language models (LLMs) are increasingly adopted in mission-critical decision-making tasks—ranging from

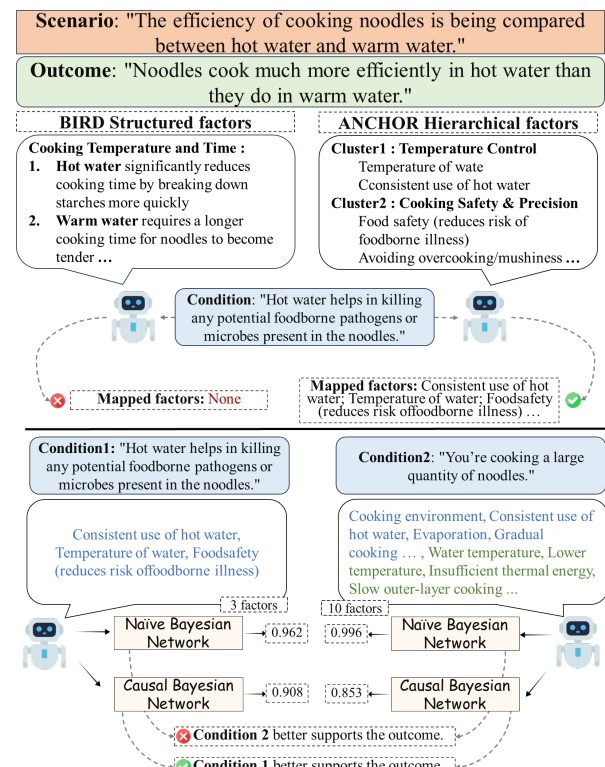

*Figure 1.* Limitations of prior abductive-Bayesian decision-making in a cooking scenario. **Top:** forward abduction produces a sparse factor space, causing 'unknown' mappings. **Bottom:** when a condition activates factors, naïve expansion adds noise and violates Naïve Bayes independence; ANCHOR mitigates both via hierarchical factor-space construction and causal Bayesian modeling.

[*]Equal contribution [1] School of Informatics, Xiamen University, Xiamen, China [2] School of Computer and Data Science, Minjiang University, Fuzhou, China [3] School of Film, Xiamen University, Xiamen, China [4] Key Laboratory of Digital Protection and Intelligent Processing of Intangible Cultural Heritage of Fujian and Taiwan, Ministry of Culture and Tourism, Xiamen University, Xiamen, China [5] Institute of Artificial Intelligence, Xiamen University, Xiamen, China . Correspondence to: Wentao Qiu <wtqiu@stu.xmu.edu.cn>, Meihong Wang <wangmh@xmu.edu.cn>, Qingqiang Wu <wuqq@xmu.edu.cn>.

*Proceedings of the 43rd International Conference on Machine Learning*, Seoul, South Korea. PMLR 306, 2026. Copyright 2026 by the author(s).

emergency response to infrastructure planning—where accurate, interpretable, and unbiased probability estimates are essential for trustworthy outcomes (McCarthy & Hayes, 1987; Sun et al., 2025). The central challenge is to reliably estimate the conditional probability $P(O_i \mid C)$ for two competing hypotheses, $O_1$ and $O_2$, given a context $C$. This context is typically composed of a high-level scenario, $S_{\text{cen}}$, and a specific downstream condition, $U$. Existing abductive frameworks build a factor space based on the general scenario $S_{\text{cen}}$, but often struggle to connect it to the specific details in $U$.

However, LLMs remain ill-suited for direct probability estimation for two key reasons: (1) they often produce numerical confidence scores that are both inaccurate and overconfident (Sun et al., 2025), and (2) they lack an out-of-the-box interpretable and controllable mechanism for explaining how these estimates are derived (Zhao et al., 2024). As a result, we cannot directly leverage LLMs' decision-making capabilities in large-scale automated scenarios, motivating the need for methods to infer more reliable probabilities from LLM outputs.

Recent methods address this by combining LLM-driven abduction with Bayesian inference. A prominent approach, exemplified by BIRD (Feng et al., 2025), as shown on the left side of Figure 1, uses *forward abduction*, where an LLM first generates a factor and then its mutually exclusive attributes. This method, however, often produces a sparse factor space, causing the mapping to be empty and leading to "unknown" predictions.

However, indiscriminately enlarging the factor space to boost coverage—as shown on the right of Figure 1—inevitably injects statistical noise and creates spurious dependencies among factors. This undermines the naïve Bayes conditional-independence assumption and distorts the resulting probability estimates (Zhang, 2004; Hand & Yu, 2001). Some works have introduced techniques such as attribute weighting, feature grouping, and structural extensions to mitigate this independence violation (Zaidi et al., 2013; Prabha et al., 2022). Other related work has explored different facets of combining LLMs and Bayesian reasoning (Reuter et al., 2025; Qiu et al., 2025; Sgouritsa et al., 2024), but the dual challenges of sparsity and dependency in abductive frameworks persist.

To overcome these limitations, we propose ANCHOR (**A**bductive **N**etwork **C**onstruction with **H**ierarchical **O**rchestration for **R**eliable probability inference), a multi-stage framework that: (1) Iteratively builds a dense factor space from LLM-generated sentences using a bottom-up abduction strategy, and organizes these factors into a two-tier hierarchy via clustering and LLM-guided theming; (2) Implements a context-aware mapping pipeline using hierarchical retrieval; and (3) Constructs both a Naïve Bayes model and a Causal Bayesian Network whose parameters are initialized with LLM-elicited priors, capturing latent dependencies between factors. Results show that ANCHOR substantially reduces "unknown" predictions and produces probability estimates that are significantly more calibrated and better aligned with human preferences than state-of-the-art baselines. In summary, our contributions are as follows:

- We design a multi-stage abduction pipeline that iteratively expands the factor space with high-quality factors, substantially reducing "unknown" predictions in downstream inference.

- We integrate a causal Bayesian network to model latent factor dependencies, enhancing probability calibration beyond the naïve Bayes assumption.

- We show that ANCHOR achieves state-of-the-art performance on preference-based pairwise evaluation while substantially reducing inference time and token usage.

## 2. Related Work

**Decomposition-Based Reasoning** Like many advanced reasoning frameworks, ANCHOR decomposes complex problems into smaller components. This builds on a rich body of work, from explicitly breaking down questions into procedural steps (Wolfson et al., 2020; Press et al., 2023) or generating faithful reasoning chains (Tafjord et al., 2022; Zhou et al., 2023), to the widely used Chain-of-Thought (CoT) prompting (Wei et al., 2022). More recent structured approaches aim for greater cognitive plausibility (Yao et al., 2023; Lin et al., 2023), maintain an explicit belief graph (Kassner et al., 2023), or model latent variables over the reasoning chain or themes (Hoffman et al., 2023; Li et al., 2025). While related, the latent variables in ANCHOR's Causal Bayesian Network represent abstract concepts rather than token sequences. The primary distinction of our method, however, is its proactive factor-space construction. Instead of decomposing reactively for each query, ANCHOR employs "bottom-up abduction" to build a comprehensive, hierarchically structured factor space. This design mitigates the issue of factor sparsity, creating a persistent and reusable reasoning structure that reduces the generation of "unknown" predictions.

**Probabilistic Inference and Uncertainty Estimations** Prior work estimates LLM uncertainty from intrinsic signals such as token probabilities (Luo et al., 2026b) or verbalized confidence (Xiong et al., 2023), which are often miscalibrated, and also uses sampling-based methods (Kuhn et al., 2023) or Bayesian prompt inference (Derakhshani et al., 2023). Another line fine-tunes models to amortize Bayesian inference or automate parts of model specification (Hu et al., 2024), but these approaches can be unreliable or computationally costly in practice. In contrast, ANCHOR performs calibration through explicit probabilistic modeling (Feng et al., 2025; Hou et al., 2024). Specifically, it orchestrates a Naïve Bayes classifier (Zhang, 2004) and a latent-augmented CBN that models dependencies between factors beyond NB's conditional-independence assumption (Prabha et al., 2022; Li et al., 2025). Within this framework, the LLM supports both parameter elicitation (Nafar et al., 2025) and dynamic structure construction for the CBN (Babakov et al., 2025). We then aggregate NB and CBN posteriors to obtain a more reliable probability estimate.

**Structured Retrieval for Grounded Reasoning** Recent work in Retrieval-Augmented Generation (RAG) (Lewis et al., 2020) has moved towards structured knowledge sources, employing graphs and hierarchies to address challenges like the "knowledge gap" between local and global context (Luo et al., 2025; Edge et al., 2024; Huang et al., 2025; Zhang et al., 2024). These studies collectively demonstrate the benefits of hierarchical and hybrid retrieval strategies. While these advanced methods primarily focus on indexing and structuring existing document corpora, AN-CHOR distinguishes itself by proactively constructing its knowledge source from the ground up. Our approach first generates a comprehensive set of reasoning factors, then organizes them into an interpretable hierarchy using established clustering techniques (Petukhova et al., 2025) and LLM-guided theming (Zhang et al., 2023; Azher et al., 2025). This process yields a purpose-built structure tailored specifically for reasoning, enabling a sophisticated retrieval process founded on high-quality, contextually-aware evidence, rather than relying on a general-purpose indexed document store.

## 3. Preliminaries

### 3.1. Problem Formulation

We formulate reliable decision-making as a contextual binary inference task. For clarity, App. A.1 summarizes the notation used throughout the main text. Given a context $C$ and two competing hypotheses, $O_1$ and $O_2$, the objective is to estimate the conditional probabilities $P(O_i \mid C)$ for $i \in \{1, 2\}$. A calibrated estimate allows a system to robustly decide which hypothesis is more plausible. Following Feng et al. (2025), we decompose $C$ into a high-level scenario $S_{\text{cen}}$ and a downstream condition $u$, enabling a neutral reasoning space based on $S_{\text{cen}}$ that mitigates biases from $u$. For completeness, we treat an "unknown" prediction as abstention: letting $\mathcal{F}^\star$ denote the factor–mapping operator (see §4.2), the system yields an unknown prediction iff $|\mathcal{F}^\star(u)| = 0$ or $\max_{i \in \{1,2\}} P(O_i \mid C) < \tau$ for a preset $\tau \in (0, 1)$.

### 3.2. Abductive-Deductive Inference Framework

Some prior work (Feng et al., 2025) implements a two-stage process. First, in a step of *forward abduction*, it generates a set of $N$ discrete factors $\mathcal{F} = \{F_1, \ldots, F_N\}$ directly from the scenario $S_{\text{cen}}$. Each factor $F_j$ has a corresponding value set $\mathcal{V}_j$. The Cartesian product of these value sets, $\mathcal{V} = \mathcal{V}_1 \times \cdots \times \mathcal{V}_N$, forms the complete information space. An element $f = (f_1, \ldots, f_N) \in \mathcal{V}$ represents one fully-specified state of the world.

Next, the framework *deductively* computes the outcome probability by marginalizing over this information space:

$$P(O_i \mid C) = \sum_{f \in \mathcal{V}} P(O_i \mid f) \, P(f \mid C).$$

Here, $P(O_i \mid f)$ is the conditional probability table (CPT) and $P(f \mid C)$ is the probability of instance $f$ given the context. To make this tractable, factors are assumed to be conditionally independent given $C$, simplifying the calculation to: $P(O_i \mid C) = \sum_{f \in \mathcal{V}} P(O_i \mid f) \prod_{j=1}^{N} P(f_j \mid C)$

## 4. Methodology

In this section, we introduce ANCHOR, a three-stage framework designed to transform raw LLM outputs into well-calibrated probabilities. As illustrated in Figure 2, the process consists of factor-space construction, context-aware mapping, and probabilistic inference.

### 4.1. Iterative Abduction for Factor–Space Expansion

#### 4.1.1. BOTTOM-UP ABDUCTION STRATEGY

We define *bottom-up abduction* as a two-stage process that inverts the traditional paradigm by decoupling factor generation from structuring. First, we iteratively generate a comprehensive set of factors, and then we cluster and theme them to build a structured factor space.

The generation stage begins with an empty set $\mathcal{F}^{(0)} = \varnothing$ and iterates until reaching a target size $N_{target}$ or exceeding $T_{\max}$ rounds: (a) **Contextual sentence generation**, where a few-shot prompt elicits $b$ diverse supporting or refuting sentences per scenario, explicitly encouraging varied reasoning chains and broad aspect coverage; and (b) **Factor harvesting and validation**, where an LLM extraction prompt identifies all distinct factors in the sentences, which are parsed into a candidate set $\Delta\mathcal{F}$, validated for semantic non-redundancy and domain relevance, and merged into the main set via $\mathcal{F}^{(t+1)} = \mathcal{F}^{(t)} \cup \Delta\mathcal{F}$. Our factor-generation procedure achieves geometric convergence in recall completeness. Moreover, under self-consistency (Wang et al., 2023), querying the LLM $m$ times and taking a majority vote yields an error probability bounded by $\exp(-2m(q - 0.5)^2)$, where $q > 0.5$ denotes the accuracy of a single vote.

#### 4.1.2. SELF-CONSISTENCY AND STRUCTURAL ORGANIZATION

Once the factor set $\mathcal{F}$ is sufficiently large, we impose a hierarchical structure through a four-stage pipeline: (i) encode each factor $f$ into $\mathbb{R}^d$ using MiniLM (Wang et al., 2020); (ii) apply UMAP to project embeddings into a lower-dimensional space, reducing noise while preserving local semantic neighborhoods (McInnes et al., 2018); (iii) run HDBSCAN to discover cohesive factor groups without pre-specifying the cluster count (McInnes et al., 2017); and (iv) prompt an LLM to assign a concise theme (e.g., *Economic Feasibility*) to each cluster and remove redundant factors,

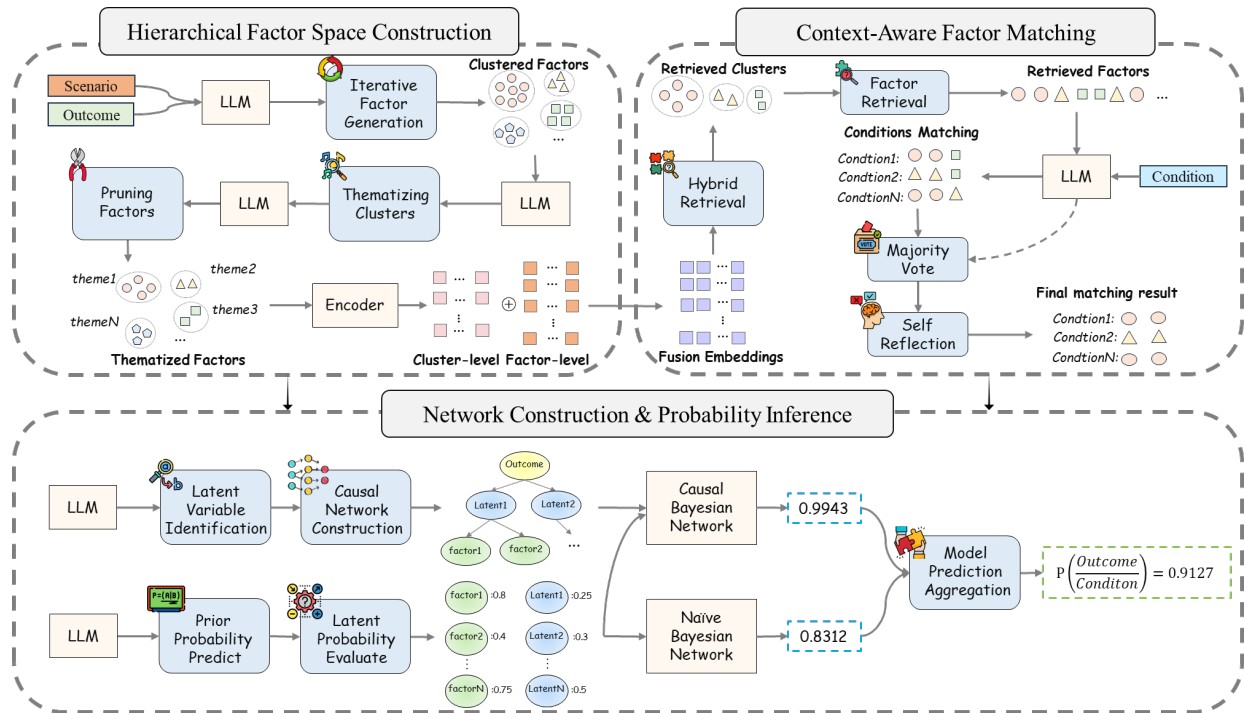

*Figure 2.* Overview of ANCHOR: (1) *Factor–Space Construction*: iterative factor generation and hierarchical clustering generate a dense, two-level factor hierarchy; (2) *Context–Aware Mapping*: perform coarse-to-fine retrieval over the factor hierarchy, then apply self-consistent filtering and reflective refinement to select factors relevant to the condition; (3) *Inference Orchestration*: construct Naïve Bayes and Causal Bayesian networks from the mapped factors—using an LLM to identify latent variables and combine their outputs into a single calibrated probability.

yielding a final hierarchical structure $\widetilde{\mathcal{F}}$. To derive factor attributes—supporting $O_1$, supporting $O_2$, or neutral—for downstream tasks, we prompt the LLM to classify each factor as "supports $O_1$", "supports $O_2$", or "neutral".

### 4.2. Context-Aware Factor Mapping

With the hierarchical factor space $\widetilde{\mathcal{F}}$ in place, mapping a downstream condition $u$ via brute-force search is computationally intractable. To address this, we propose a multi-stage pipeline that defines a mapping operator $\mathcal{F}^\star$: starting from a broad retrieval stage to maximize recall, then applying targeted filtering and precision-driven refinement, it maps a condition $u$ to a compact, high-confidence factor set $\mathcal{F}^\star(u)$.

#### 4.2.1. HIERARCHICAL RETRIEVAL FOR CANDIDATE GENERATION

Our retrieval design leverages the two-level structure of the factor space $\widetilde{\mathcal{F}}$ to efficiently generate a high-recall candidate set, $\mathcal{F}_{\text{cand}}(u)$. The process begins by embedding the condition $u$ and all factor cluster prototypes into a shared vector space $\mathbb{R}^d$. A key design element is the cluster prototype itself, formulated as a weighted average of its thematic label

and its member factor embeddings:

$$e_{C_j} = \alpha \cdot e_{\text{theme}} + (1 - \alpha) \cdot \frac{1}{|F_j|} \sum_{f \in F_j} e_f \qquad (1)$$

where $e$ denotes an embedding vector and $\alpha \in [0, 1]$ is a weighting parameter. This formulation balances high-level categorical meaning with fine-grained distributional semantics.

We then perform a **coarse-to-fine search** using K-Nearest Neighbors (KNN) (Guo et al., 2003) to rapidly zero in on relevant factors:

1. **Coarse Search (Cluster-Level):** We first identify the top-$K_1$ clusters whose prototypes are most semantically similar to the condition embedding $e_u$. This step acts as a high-level filter, dramatically narrowing the search space to the most promising regions.

2. **Fine Search (Factor-Level):** Within this curated set of clusters, we then conduct a more granular search, retrieving the top-$K_2$ individual factors from each selected cluster that are closest to $e_u$.

The union of factors retrieved from this two-step process forms the candidate set $\mathcal{F}_{\text{cand}}(u)$, which is intentionally

broad to ensure no relevant factors are prematurely discarded.

### 4.2.2. ROBUST FACTOR SELECTION VIA SELF-CONSISTENCY AND REFLECTION

Our high-recall retrieval can introduce noisy candidates, so we apply a two-stage selection procedure. First, following the self-consistency principle (Wang et al., 2023), we query an LLM $R$ times with the same prompt to select the subset of $\mathcal{F}_{\text{cand}}(u)$ that is directly supported by the condition $u$, and tally per-factor votes $v_f(u) = \sum_{r=1}^{R} \mathbf{1}[f \in m^{(r)}(u)]$, where $m^{(r)}(u)$ is the subset returned in the $r$-th query. We keep factors whose votes exceed a configurable threshold $\gamma$ (e.g., $\gamma = \lceil \text{vote\_ratio} \cdot R \rceil$), producing a filtered set $\mathcal{F}_{\text{vote}}(u)$ that prunes spurious or weakly supported factors. Second, as a final precision check, we pass $\mathcal{F}_{\text{vote}}(u)$ to a reflection prompt that explicitly asks the LLM to remove any remaining factors lacking clear, direct relevance to $u$; such structured self-critique is commonly used to reduce hallucinations and improve factual precision (Ji et al., 2023; Renze & Guven, 2024), yielding the final factor set $\mathcal{F}^\star(u)$.

### 4.3. Probabilistic Inference with Elicited Parameters

#### 4.3.1. PARAMETERIZATION AND INFERENCE

The final stage of our framework transforms the mapped factor set, $\mathcal{F}^\star(u)$, into a calibrated probability for the competing hypotheses, $O_1$ and $O_2$. To achieve this, we first define the structure of two probabilistic models—a Naïve Bayes (NB) model and a Causal Bayesian Network (CBN). We then describe the process of eliciting the necessary parameters from an LLM and, finally, how these parameters are used to perform inference in each model.

**Naïve Bayes Model.** The Naïve Bayes model is structured on the assumption that all factors are conditionally independent given the hypothesis. It consists of a root Outcome node (representing $O_1$ and $O_2$) with directed edges to each child factor node $f_j \in \mathcal{F}^\star(u)$.

**Latent-Augmented Causal Bayesian Network.** To capture dependencies between factors, we construct a CBN whose structure is learned dynamically for each scenario. We prompt an LLM to act as a causal discovery engine: given the list of relevant factors, it identifies a set of latent variables $\mathcal{L} = \{L_1, \ldots, L_k\}$ and partitions the factors among them. Each latent variable $L_i \in \{0, 1\}$ represents an abstract concept that mediates correlations among the factors assigned to it. Accordingly, we add directed edges Outcome $\to L_i$ and $L_i \to f_j$ for each factor $f_j$ grouped under $L_i$. Outcomes serve as roots and the latents are intermediate parents shared by factors, relaxing NB's independence assumption by making factors independent only when conditioned on their latent parent.

Inspired by prior work validating the use of LLMs to elicit informative priors (Riegler et al., 2025; Nafar et al., 2025; Capstick et al., 2025), our framework efficiently parameterizes its Bayesian networks. We directly query the LLM to obtain the necessary conditional probabilities—both at the factor and latent levels—thus bypassing the need for costly data sampling.

**Factor-Level Parameters**: For each factor $f$, we elicit its posterior probability given hypothesis $O_1$, denoted as $\phi_f = P(O_1 \mid f)$. This captures the evidential strength of a single factor in favor of $O_1$. For the Naïve Bayes (NB) model, which is parameterized by likelihoods $P(f \mid O_k)$, we adopt a simple binary, symmetric-prior approximation and set $\theta_f = P(f \mid O_1) \approx \phi_f, P(f \mid O_2) \approx 1 - \phi_f$.

**Latent-Level Parameters**: Exclusively for the Causal Bayesian Network (CBN), we elicit the conditional probability of each latent variable $L_i$ given a hypothesis, such as $P(L_i = 1 \mid O_1)$ and $P(L_i = 0 \mid O_2)$. To ensure numerical stability, all elicited probabilities are smoothed. These parameters are then used to build and perform inference in our two probabilistic models.

**Evidence representation.** We treat each factor $f \in \mathcal{F}^\star(u)$ as a binary indicator, where $f = 1$ denotes that the factor is active for $u$. Let $\mathcal{E} \subseteq \mathcal{F}^\star(u)$ denote the set of observed active factors. For simplicity, factors not in $\mathcal{E}$ are treated as unobserved and are omitted from the likelihood in both NB and CBN.

The NB model is parameterized by using the elicited posterior $\phi_f$ as a proxy for the likelihood, defining the CPT for each factor $f$ as $P(f \mid O_1) = \theta_f \approx \phi_f, P(f \mid O_2) \approx 1 - \phi_f$. Given an evidence set $\mathcal{E}$ and assuming a uniform prior over outcomes, the posterior probability for hypothesis $O_1$ is calculated as:

$$P(O_1 \mid \mathcal{E}) = \frac{\prod_{f \in \mathcal{E}} P(f \mid O_1)}{\prod_{f \in \mathcal{E}} P(f \mid O_1) + \prod_{f \in \mathcal{E}} P(f \mid O_2)}. \quad (2)$$

The CBN's CPTs are parameterized as follows. **Latent Nodes** $(L_i)$ are children of the Outcome node and are parameterized by the elicited conditional probabilities $P(L_i = 1 \mid \text{Outcome} = O_k)$ and $P(L_i = 0 \mid \text{Outcome} = O_k) = 1 - P(L_i = 1 \mid \text{Outcome} = O_k)$. The CPT for a **Factor Node** $(f_j)$, conditioned on its latent parent $L_i$, uses the same NB-style likelihood parameters, specifically $P(f_j \mid L_i = 1) = \theta_{f_j} \approx \phi_{f_j}, P(f_j \mid L_i = 0) \approx 1 - \phi_{f_j}$. The CPT for the **Outcome Node**, $P(\text{Outcome} \mid L_1, \ldots, L_k)$, is derived from the elicited latent-level parameters $P(L_i \mid O_k)$ via Bayes' rule. **Proxy-likelihood and CBN inference.** We use the elicited posterior $\phi_f = P(O_1 \mid f = 1)$ as a monotone proxy for the NB likelihood $\theta_f \coloneqq P(f = 1 \mid O_1)$, and analyze the sensitivity of this approximation in Section A.2.3.

For the CBN with structure Outcome $\rightarrow L_i \rightarrow f_j$, we compute $P(\text{Outcome} = O_1 \mid \mathcal{E})$ by exact marginalization over $\mathcal{L}$ using variable elimination (Zhang & Poole, 1994).

### 4.3.2. MODEL AGGREGATION

To synthesize the predictions from our NB and CBN models into a single, more reliable estimate, we employ the Linear Opinion Pool (LOP) strategies: The LOP is a simple and powerful method that forms a weighted average of individual model predictions (Stone, 1961; Neyman & Roughgarden, 2023; Stratigakos et al., 2025). We select it as our primary strategy for its computational efficiency and proven robustness. The aggregated probability is a convex combination: $P_{\text{LOP}}(O_1 \mid \mathcal{E}) = \sum_{M \in \{\text{NB,CBN}\}} w_M \cdot P(O_1 \mid \mathcal{E}, M)$, where the weights $w_M$ are fixed and sum to 1.

## 5. Experiment

### 5.1. Experimental Setups

**Datasets** We evaluate our model on three reasoning and planning benchmarks from Feng et al. (2025), each formatted as a decision-support tuple $(S_{\text{cen}}, U, O)$: **Today** (1,000 instances), **Plasma** (279 scenarios, 1,395 instances) and **Common2Sense** (216 scenarios, 3,822 instances), the Common2Sense's test split being manually expanded under an unbiased protocol to boost diversity. To assess decision-making beyond planning, we further sample 100 examples from each of the four fact-checking datasets—**XSum** (Tang et al., 2024), **CNN** (Tang et al., 2024), **ExpertQA** (Malaviya et al., 2024) and **COVID** (Saakyan et al., 2021)—treating each document as condition $U$ and each claim as outcome $O$, with $S_{\text{cen}}$ left empty. (dataset details refer to B.3).

**Settings** We conduct all experiments with four LLMs: Qwen2.5-32B, Qwen2.5-72B (Bai et al., 2023), DeepSeekV3-671B (DeepSeek-AI, 2024), and GPT-4 (OpenAI, 2023). The Qwen models were deployed locally, while DeepSeekV3-671B was accessed through DeepSeek's official API. We set $K_1 = 3$ and $K_2 = 5$ in the KNN search; more hyperparameters and settings are provided in Appendix B.4. All experiments run on 4× NVIDIA RTX 4090 (24GB) GPUs.

**Preference-based Pairwise Evaluation** To begin, we evaluate our model on Common2Sense using the *preference-based pairwise evaluation* framework introduced by Feng et al. (2025). The experimental setup is as follows. Given a scenario $S_{\text{cen}}$ and two potential outcomes, $O_1$ and $O_2$, we select two distinct additional conditions, $U_1$ and $U_2$. Crucially, both conditions are constructed to support the same outcome, $O_1$, over $O_2$. These are then used to form two slightly different contexts: $C_1 = S_{\text{cen}} + U_1$ and $C_2 = S_{\text{cen}} + U_2$. The

model's task is to assign conditional probabilities $P(O_i|C_1)$ and $P(O_i|C_2)$ for each outcome $i \in \{1, 2\}$. A successful evaluation requires the model to correctly discern the subtle difference in the degree of support between the two contexts. For instance, if human judgment determines that $C_1$ provides stronger support for $O_1$ than $C_2$ does, we expect the model's probability assignments to satisfy the following relationship: $P(O_1|C_1) > P(O_1|C_2) > P(O_2|C_2) > P(O_2|C_1)$, we set the abstention threshold $\tau = 0$, so the system yields an unknown prediction only when no factors are matched, i.e., $|\mathcal{F}^\star(u)| = 0$.

**Decision Making Evaluation** We evaluate ANCHOR's decision-making on reasoning and planning datasets (Plasma, Today) and four fact-checking datasets for generalization (ExpertQA, XSum, COVID, CNN). In each task, given a context $C$ and two outcomes $O_1$ and $O_2$, the model must select the more plausible one. A decision is considered correct if the chosen outcome has a higher conditional probability, e.g., satisfying $P(O_1|C) > P(O_2|C)$ when $O_1$ is the ground truth.

**Baseline Methods** We compare our approach against several baselines for probability estimation from LLMs in our experiments: (1) **BIRD**: Builds a structured factor space via single-pass abductive reasoning and uses a Naïve Bayes model for inference (Feng et al., 2025). (2) **Vanilla**: Prompts the model to verbalize its estimated probability (Wang et al., 2023). (3) **Logits**: Converts the normalized token probability of the decision token into a probability score. (4) **CoT**: Elicits a chain-of-thought reasoning process before asking for the final probability (Wei et al., 2022). (5) **Compare**: Selects which of two conditions better supports the outcome without producing individual probabilities, an unfair setting for methods relying on those estimates (Feng et al., 2025). (6) **Factor-based**: Averages probabilities over five factors generated with knowledge of the gold outcome, making direct comparison invalid.

### 5.2. Results

The results are listed in Table 1. Our framework, ANCHOR, exhibits a superior alignment with human preference, achieving a top F1 of 60.6%. This demonstrates that ANCHOR produces reliable probability estimations.

Some baselines, such as Compare, perform joint context processing, giving them an unfair advantage. We also introduce Factor-based, another unfair baseline that assumes the model can construct a perfect factor space and always match the gold-aligned factor. Despite these advantages, ANCHOR still outperforms both baselines—achieving up to 5% in F1. Even our smaller Qwen2.5-32B model outperforms the baselines built on the larger Qwen2.5-72B. In terms of coverage, ANCHOR achieves near-complete coverage with its

*Table 1.* Preference-based pairwise evaluation using F1 score on the Common2sense dataset. Context1 is the F1 score for the class where the first context better supports the outcome than the second, and Context2 is for the reverse. Same is the F1 score for cases where both contexts offer equal support. Average is the global micro-averaged F1 score. For BIRD[‡] and ANCHOR, reported numbers are mean with standard deviation over 5 runs. BIRD[†] uses the original, unmodified mapping output of the BIRD method. BIRD[‡] employs a relaxed mapping condition, forcing the model to match each condition with at least one factor from the factor space. Unlike BIRD, which manually filtered "unknown" cases, we remove this intervention for fairness. Best results are bolded, and second-best results are underlined.

| Method | Model | Context1 | Context2 | Same | Average | Coverage |
|---|---|---|---|---|---|---|
| Random | Guess | 0.333 | 0.333 | 0.333 | 0.333 | – |
| CoT | Qwen2.5-32B | 0.394 | 0.370 | 0.118 | 0.319 | – |
| | Qwen2.5-72B | 0.370 | 0.382 | 0.159 | 0.311 | – |
| | DeepSeek-V3-671B | 0.414 | 0.395 | 0.146 | 0.338 | – |
| Vanilla | Qwen2.5-32B | 0.528 | 0.528 | 0.211 | 0.462 | – |
| | Qwen2.5-72B | 0.526 | 0.528 | 0.183 | 0.470 | – |
| | DeepSeek-V3-671B | 0.489 | 0.498 | 0.137 | 0.425 | – |
| Logits | Qwen2.5-32B | 0.504 | 0.491 | 0.150 | 0.445 | – |
| | Qwen2.5-72B | 0.523 | 0.500 | 0.156 | 0.447 | – |
| | DeepSeek-V3-671B | 0.516 | 0.513 | 0.162 | 0.453 | – |
| Compare | Qwen2.5-32B | 0.583 | 0.480 | 0.221 | 0.477 | – |
| | Qwen2.5-72B | 0.566 | 0.570 | 0.267 | 0.534 | – |
| | DeepSeek-V3-671B | 0.560 | 0.578 | 0.286 | 0.542 | – |
| | GPT-4 | 0.587 | 0.548 | 0.302 | 0.556 | – |
| Factor-based | Qwen2.5-32B | 0.556 | 0.564 | 0.054 | 0.525 | – |
| | Qwen2.5-72B | 0.509 | 0.509 | 0.250 | 0.496 | – |
| | DeepSeek-V3-671B | 0.553 | 0.546 | 0.165 | 0.506 | – |
| BIRD | Qwen2.5-72B[†] | 0.497 | 0.412 | 0.271 | 0.421 | 87.77% |
| | Qwen2.5-72B[‡] | $0.568_{\pm 0.012}$ | $0.523_{\pm 0.014}$ | $\mathbf{0.306}_{\pm 0.011}$ | $0.526_{\pm 0.013}$ | 99.99% |
| ANCHOR (ours) | Qwen2.5-32B | $\underline{0.602}_{\pm 0.012}$ | $0.583_{\pm 0.012}$ | $0.146_{\pm 0.018}$ | $\underline{0.563}_{\pm 0.017}$ | 99.79% |
| | Qwen2.5-72B | $\mathbf{0.610}_{\pm 0.021}$ | $\mathbf{0.611}_{\pm 0.035}$ | $0.278_{\pm 0.014}$ | $\mathbf{0.606}_{\pm 0.030}$ | 99.95% |

default mapping—no manual intervention required—while BIRD attains only 87.8% under its original mapping and must relax matching to reach comparable coverage.

ANCHOR is designed to be highly sensitive to nuanced differences between contexts, which results in its F1 being lower on the Same class compared to other categories. Given the severe class imbalance (the Same class is only 9.82% of the dataset), the Micro Avg F1 score provides a more meaningful measure of overall performance, where our method clearly excels.

For decision-making, following Feng et al. (2025), when the BIRD baseline predicts "unknown", we default to the CoT method to ensure a decision is made. As other methods do not predict "unknown", we omit the coverage metric. ExpertQA, XSum, CNN, and Today represent more challenging settings due to long contexts or temporal reasoning, while COVID and Plasma are relatively simpler (Feng et al., 2025). For decision-making on fact-checking datasets, we

convert probabilistic outputs into support labels by applying a fixed threshold, following prior work in fact verification (Jayaweera et al., 2024). For the Plasma and Today datasets, we adhere to the evaluation of (Feng et al., 2025). As shown in Table 2, ANCHOR consistently outperforms all baselines at the 72B scale and even surpasses the 671B model on the harder tasks.

### 5.3. Ablation Study

To validate our design, we conducted an ablation study on Common2Sense (Table 3), analyzing five key variants. In addition to ablating core components, the study compares our standard LOP with an alternative aggregation strategy: Bayesian Model Averaging (BMA), a theoretically rigorous method that weights models by their posterior probabilities (Hoeting et al., 1999; Fragoso et al., 2018).

The five variants are: (1) **w/o cbn**, which uses only the Naïve Bayes model in the aggregation step; (2) **w/o nb**, which uses

*Table 2.* Balanced accuracy on fact-checking tasks and accuracy on reasoning/planning tasks. Methods marked ⋆ make direct decisions without probability outputs. For BIRD and ANCHOR, we report mean ± std over 5 runs.

| Method | ExpertQA | XSum | COVID | CNN | Today | Plasma |
|---|---|---|---|---|---|---|
| | | | DeepSeek-V3-671B | | | |
| Vanilla | 0.538 | 0.537 | 0.752 | 0.495 | 0.710 | 0.774 |
| Vanilla⋆ | 0.562 | 0.483 | 0.686 | 0.500 | 0.647 | 0.781 |
| CoT | 0.541 | 0.527 | 0.685 | 0.495 | 0.783 | 0.784 |
| CoT⋆ | 0.498 | 0.517 | **0.754** | 0.462 | 0.803 | **0.850** |
| | | | Qwen2.5-72B | | | |
| Vanilla | 0.517 | 0.557 | 0.538 | 0.604 | 0.583 | 0.604 |
| Vanilla⋆ | 0.530 | 0.473 | 0.686 | 0.500 | 0.597 | 0.649 |
| CoT | 0.590 | 0.531 | 0.721 | 0.589 | 0.593 | 0.565 |
| CoT⋆ | 0.552 | 0.484 | 0.701 | 0.500 | 0.713 | 0.745 |
| BIRD | 0.582 | 0.511 | 0.664 | 0.540 | 0.754 | 0.718 |
| | ± 0.023 | ± 0.022 | ± 0.020 | ± 0.027 | ± 0.035 | ± 0.022 |
| ANCHOR (ours) | **0.611** | **0.564** | 0.726 | **0.621** | **0.817** | 0.764 |
| | ± 0.019 | ± 0.016 | ± 0.013 | ± 0.029 | ± 0.031 | ± 0.022 |

only the Causal Bayesian Network model in the aggregation step; (3) **w/o cluster**, which expands factors without hierarchical clustering, equivalent to simply increasing the number of factors; (4) **w/o hierarchy**, which retains clustering but removes weighted fusion and coarse-to-fine retrieval; (5) **w/o pe-llm**, which replaces LLM-elicited parameters with frequency-based estimates (B.1); and (6) **ANCHOR (BMA)**, which implements the BMA strategy (B.2).

*Table 3.* Ablation results of ANCHOR. The full model is highlighted. Red and green subscripts denote performance decreases and increases relative to the full model.

| Component | Ctx. 1 | Ctx. 2 | Same | Avg. | Cov. |
|---|---|---|---|---|---|
| | | | Qwen2.5-32B | | |
| w/o cbn | 0.592 | 0.579 | 0.152 | 0.558 ↓0.014 | – |
| w/o nb | 0.518 | 0.514 | 0.197 | 0.479 ↓0.093 | – |
| w/o cluster | 0.480 | 0.497 | 0.193 | 0.457 ↓0.115 | 98.93% |
| w/o hierarchy | 0.514 | 0.511 | 0.131 | 0.491 ↓0.081 | 99.84% |
| w/o pe-llm | 0.459 | 0.484 | 0.271 | 0.435 ↓0.137 | – |
| ANCHOR (BMA) | 0.615 | 0.592 | 0.152 | 0.575 ↑0.003 | – |
| ANCHOR (full) | **0.612** | **0.587** | 0.152 | **0.572** | 99.79% |
| | | | Qwen2.5-72B | | |
| w/o cbn | 0.530 | 0.518 | 0.219 | 0.506 ↓0.090 | – |
| w/o nb | 0.592 | 0.583 | 0.260 | 0.557 ↓0.039 | – |
| w/o cluster | 0.420 | 0.429 | 0.173 | 0.400 ↓0.196 | 98.82% |
| w/o hierarchy | 0.515 | 0.512 | 0.127 | 0.491 ↓0.105 | 99.92% |
| w/o pe-llm | 0.505 | 0.549 | 0.281 | 0.483 ↓0.113 | – |
| ANCHOR (BMA) | 0.604 | 0.621 | 0.281 | 0.592 ↓0.004 | – |
| ANCHOR (full) | **0.626** | **0.627** | 0.281 | **0.596** | 99.95% |

The "–" in the Coverage column indicates that the variant did not affect coverage. While coverage remains near 100% for most variants, the dramatic drop in (2) confirms that

unstructured factor expansion is insufficient. The performance declines in (4) and (3) underscore the importance of accurate parameter elicitation and our two-level hierarchy, respectively. Finally, both aggregation strategies—LOP and BMA—achieve similarly high coverage and comparable F1 scores, demonstrating that effective fusion, combined with hierarchical mapping and precise elicitation, is key to ANCHOR's robust performance.

## 5.4. Analysis

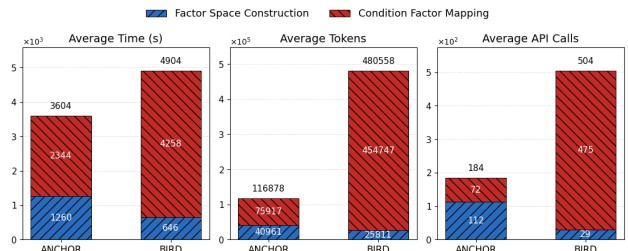

*(a)* Cost analysis of ANCHOR vs. BIRD: average per-scenario runtime, token usage, and API calls by stage.

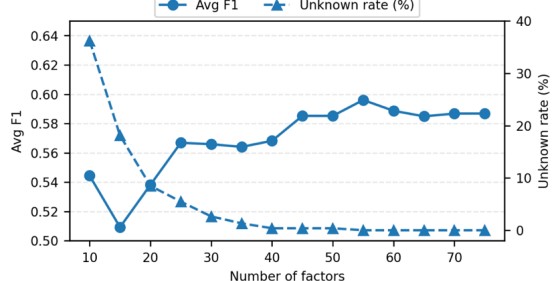

*(b)* Unknown rate and F1 scores for ANCHOR as the number of factors increases.

*Figure 3.* Cost and coverage–accuracy analysis for ANCHOR and BIRD.

*Table 4.* BIRD‡ variants on ANCHOR's factor space and mapping on Common2sense. Each "+" denotes an incremental add-on to the configuration in the row above.

| Setting | Avg. | Cov. | Tok./B | Time/B |
|---|---|---|---|---|
| BIRD‡ baseline | 0.513 | 99.99% | 1.00 | 1.00 |
| + ANCHOR-generated factors | 0.532 | 99.12% | 2.98 | 2.87 |
| + ANCHOR mapping | 0.526 | 98.81% | 0.17 | 0.55 |
| + LOP (Naïve Bayes + CBN) | **0.568** | 98.81% | 0.17 | 0.55 |

Figure 3 reports the computation time, token usage, and API calls on the Common2Sense dataset, using Qwen2.5-72B under the same experimental setup as the main results in Table 1. On average, per scenario, ANCHOR runs in about 0.74× the time, uses 0.24× the tokens, and makes 0.37× the API calls of BIRD on this benchmark. While ANCHOR invests more in factor-space construction, its hierarchical

condition–factor mapping significantly reduces downstream cost. For ANCHOR, as more factors are added, the unknown rate falls and the average F1 generally improves, reflecting a trade-off between coverage and accuracy. Appendix B.6 further details these trade-offs, including retrieval-method ablations, sensitivity to $K_1/K_2$, and a calibration study in App. B.5.2 showing that performance remains largely insensitive to the probability-estimator LLM, provided that it supplies coarse discriminative signals. We provide detailed case studies for factor spaces and mappings in App. C.1, and include the prompts used in our experiments in App. C.3.

On Common2sense with Qwen2.5-72B, we transplant the original BIRD[‡] pipeline into ANCHOR's factor space while retaining BIRD-style CPT training, and compare four variants: the BIRD[‡] baseline, BIRD[‡] with ANCHOR-generated factors, ANCHOR factors plus context-aware mapping, and the latter further aggregated with a CBN and Naïve Bayes via a linear opinion pool. Table 4 reports Avg (global micro-averaged F1 over CONTEXT1, CONTEXT2, and SAME), coverage, and mapping-stage cost normalized by the BIRD[‡] baseline (Tok./B, Time/B). Replacing BIRD[‡] factors with ANCHOR's denser factor space improves Avg from $0.513$ to $0.532$ but increases mapping cost by nearly $3\times$, whereas applying ANCHOR's mapping restores efficiency while still outperforming the baseline; adding CBN aggregation yields the best Avg ($0.568$) at essentially the same reduced cost, showing that factor-space construction, mapping, and CBN aggregation each contribute beyond a stronger factor generator alone (details in App. B.7).

## 6. Conclusion

We introduced the ANCHOR framework to generate reliable and calibrated probability estimates from Large Language Models for critical decision-making. By integrating a hierarchical factor-space construction with the orchestration of Naïve Bayes and a Causal Bayesian Network, our framework showed notable enhancements in decision-making accuracy and alignment with human judgment. Our experimental results underscore the effectiveness of ANCHOR, thereby increasing the practical utility of LLMs in high-stakes scenarios and fostering more trustworthy autonomous systems.

## Acknowledgements

We thank the anonymous reviewers and area chairs for their constructive comments and suggestions. This work was supported by the Noncommunicable Chronic Diseases-National Science and Technology Major Project(No. 2023ZD0509703), the Solfeggio Ear-Training Intelligent Robot and Cloud Platform RD Project for Music Education (No. 2024CXY0102), the 3D Visualization Digital Twin Integrated Control System Project(No. 2023CXY0111), the Pre-research Project for Introduced Talents of Minjiang University (No. MJY25025), the Public Technology Service Platform Project of Xiamen City(No. 3502Z20231043), and the Fujian Provincial Science and Technology Major Project (No. 2024HZ022003).

## Impact Statement

This work improves decision support for contextual binary inference by combining LLM-generated reasoning factors with explicit Bayesian inference, producing more calibrated probabilities and a traceable audit trail while reducing abstentions (*unknown*) and lowering inference time and token cost. However, because the factor hierarchy, latent grouping, and elicited parameters depend on LLM behavior, biases or hallucinated structures may be propagated into the resulting Bayesian models, and reducing *unknown* can trade abstention for confident but incorrect factor mappings under distribution shift or adversarial inputs. We therefore position ANCHOR as a decision-support tool rather than a fully automated decision-maker, and recommend human oversight, conservative abstention thresholds, and routine audits of mapped factors and inferred structures, with robustness, fairness, and adversarial resilience treated as prerequisites for high-stakes deployment.

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

# A. Appendix A

## A.1. Notation

Table 5 summarizes the notation used in the main text; although some symbols may be overloaded in later appendix derivations and proofs under local conventions, their meanings are explicitly specified in context and remain internally consistent.

*Table 5.* Notation used throughout the paper.

| Symbol | Type / Range | Meaning |
|---|---|---|
| $C$ | context | Full context for inference. |
| $S_{\text{cen}}$ | scenario | High-level scenario (neutral reasoning space). |
| $U$ / $u$ | condition | Downstream condition; $u$ is a concrete instance. |
| $O_i$ | $i \in \{1, 2\}$ | Outcome indexed by $i$. |
| $P(\cdot)$ | prob. | Probability measure. |
| $P(O_i \mid C)$ | $[0, 1]$ | Target conditional probability of $O_i$ given $C$. |
| $\tau$ | $(0, 1)$ | Abstention threshold for "unknown". |
| $\mathcal{F}$ | set | Set of discrete factors. |
| $N$ | $\mathbb{N}$ | Number of baseline factors: $\mathcal{F} = \{F_1, \ldots, F_N\}$. |
| $F_j$ | factor | $j$-th baseline factor. |
| $\mathcal{V}_j$ | set | Value set (attributes) of $F_j$. |
| $\mathcal{V}$ | set | Information space $\mathcal{V}_1 \times \cdots \times \mathcal{V}_N$. |
| $f = (f_1, \ldots, f_N)$ | state | Fully specified world state in $\mathcal{V}$. |
| $f_j$ | entry | $j$-th component in $f$. |
| $P(O_i \mid f)$ | CPT | CPT value for $O_i$ given state $f$. |
| $P(f \mid C)$ | $[0, 1]$ | Probability of state $f$ under $C$. |
| $\mathcal{F}^{\star}$ | operator | Factor-mapping operator. |
| $\mathcal{F}^{\star}(u)$ | set | Final mapped factors for condition $u$. |
| $\|\mathcal{F}^{\star}(u)\|$ | $\mathbb{N}$ | Size of mapped set |
| $\mathcal{F}^{(t)}$ | set | Factor set after round $t$ in iterative abduction. |
| $\mathcal{F}^{(0)} = \varnothing$ | set | Empty initialization. |
| $t$ | index | Iteration index. |
| $N_{target}$ | $\mathbb{N}$ | Target factor-set size. |
| $T_{\max}$ | $\mathbb{N}$ | Maximum abduction rounds. |
| $b$ | $\mathbb{N}$ | Generated sentences per scenario per round. |
| $\Delta \mathcal{F}$ | set | Newly harvested candidate factors. |
| $m$ | $\mathbb{N}$ | Self-consistency sample size (votes). |
| $q$ | $(0.5, 1]$ | Single-vote accuracy (assumed $> 0.5$). |
| $\widetilde{\mathcal{F}}$ | hierarchy | Final two-level hierarchical factor space. |
| $d$ | $\mathbb{N}$ | Embedding dimension; $\mathbb{R}^d$ is embedding space. |
| $\mathcal{F}_{\text{cand}}(u)$ | set | High-recall candidate factors retrieved for $u$. |
| $e_x$ | vector | Embedding of object $x$ (e.g., $e_u, e_f, e_{\text{theme}}$). |
| $e_{C_j}$ | vector | Prototype embedding of cluster $C_j$. |
| $e_{\text{theme}}$ | vector | Embedding of a cluster theme label. |
| $F_j$ (members) | set | Member-factor set of cluster $C_j$; $\|F_j\|$ is its size. |
| $\alpha$ | $[0, 1]$ | Theme vs. member-mean mixing weight in $e_{C_j}$. |
| $K_1, K_2$ | $\mathbb{N}$ | Top clusters (coarse) / top factors per cluster (fine). |
| $R$ | $\mathbb{N}$ | Number of repeated LLM selections for voting. |
| $m^{(r)}(u)$ | set | Selected subset returned by the $r$-th query. |
| $v_f(u)$ | $\{0, \ldots, R\}$ | Vote count for factor $f$. |
| $\gamma$ | $\{0, \ldots, R\}$ | Vote threshold. |
| $\mathcal{F}_{\text{vote}}(u)$ | set | Vote-filtered factor set. |
| $\mathcal{L} = \{L_1, \ldots, L_k\}$ | set | Latent variables in CBN; $L_i \in \{0, 1\}$. |
| $\phi_f$ | $[0, 1]$ | Elicited posterior strength: $\phi_f = P(O_1 \mid f)$. |
| $\theta_f$ | $[0, 1]$ | Proxy likelihood in NB: $\theta_f = P(f \mid O_1) \approx \phi_f$. |
| $\mathcal{E}$ | set | Positive evidence (active factors), $\mathcal{E} \subseteq \mathcal{F}^{\star}(u)$. |
| $P_{\text{LOP}}(O_1 \mid \mathcal{E})$ | $[0, 1]$ | Aggregated posterior probability. |
| $M$ | $\in \{\text{NB, CBN}\}$ | Model index. |
| $w_M$ | $[0, 1]$ | Fixed mixture weight, $\sum_M w_M = 1$. |
| $P(O_1 \mid \mathcal{E}, M)$ | $[0, 1]$ | Posterior under model $M$. |

**A.2. Theoretical Analysis of the Causal Bayesian Network (CBN)**

A.2.1. MODEL DEFINITION AND FACTORIZATION

Let Outcome $\in \{O_1, O_2\}$ denote the binary outcome node, and let the discovered latent set be $\mathcal{L} = \{L_1, \ldots, L_k\}$ with $L_i \in \{0, 1\}$. Each factor $f_j$ is assigned to exactly one latent (parent) by the LLM-discovered partition; denote this parent by $\pi(j) \in \{1, \ldots, k\}$ and let $\mathcal{F}_i = \{f_j : \pi(j) = i\}$ be the factor group of $L_i$.

Following the main text, we assume a uniform prior over outcomes, i.e., $P(\text{Outcome} = O_1) = P(\text{Outcome} = O_2) = \frac{1}{2}$. We elicit latent-level parameters

$$\alpha_i^{(1)} := P(L_i = 1 \mid O_1), \qquad \alpha_i^{(2)} := P(L_i = 1 \mid O_2),$$

and use the NB-style factor likelihood parameters

$$P(f_j = 1 \mid L_{\pi(j)} = 1) = \theta_{f_j} \approx \phi_{f_j}, \qquad P(f_j = 1 \mid L_{\pi(j)} = 0) = 1 - \theta_{f_j}.$$

(We assume all elicited probabilities are smoothed so that $\theta_{f_j} \in [\varepsilon, 1 - \varepsilon]$ for some $\varepsilon \in (0, \frac{1}{2})$.)

Under the directed structure Outcome $\to L_i \to f_j$, the joint distribution factorizes as

$$P(\text{Outcome}, \mathcal{L}, \mathcal{F}) = P(\text{Outcome}) \prod_{i=1}^{k} P(L_i \mid \text{Outcome}) \prod_{j=1}^{|\mathcal{F}|} P(f_j \mid L_{\pi(j)}). \tag{3}$$

This implies conditional independence of factors given their latent parent, and conditional independence of latents given Outcome.

A.2.2. CLOSED-FORM LIKELIHOOD AND POSTERIOR FOR CBN

We interpret the evidence set $\mathcal{E}$ exactly as in the main text: $\mathcal{E}$ is the set of factors observed as present/true (i.e., $f = 1$ for all $f \in \mathcal{E}$), while all other factors are unobserved and marginalized out.

**Theorem A.1** (CBN evidence likelihood decomposes by latent groups). *Let $\mathcal{E}_i := \mathcal{E} \cap \mathcal{F}_i$ denote the subset of observed factors assigned to $L_i$. Then for $r \in \{1, 2\}$,*

$$P(\mathcal{E} \mid O_r) = \prod_{i=1}^{|\mathcal{L}|} \left( P(L_i = 1 \mid O_r) \prod_{f \in \mathcal{E}_i} P(f = 1 \mid L_i = 1) + P(L_i = 0 \mid O_r) \prod_{f \in \mathcal{E}_i} P(f = 1 \mid L_i = 0) \right). \tag{4}$$

*Consequently, the CBN posterior admits the closed form*

$$P(\text{Outcome} = O_1 \mid \mathcal{E}) = \frac{P(\mathcal{E} \mid O_1)}{P(\mathcal{E} \mid O_1) + P(\mathcal{E} \mid O_2)}. \tag{5}$$

**Proof of Theorem A.1.** Starting from (3) and conditioning on Outcome $= O_r$,

$$P(\mathcal{E} \mid O_r) = \sum_{\ell \in \{0,1\}^k} \sum_{\mathcal{F} \backslash \mathcal{E}} \left( \prod_{i=1}^{k} P(L_i = \ell_i \mid O_r) \right) \left( \prod_{j} P(f_j \mid L_{\pi(j)} = \ell_{\pi(j)}) \right),$$

where the inner summation marginalizes all unobserved factors. Because for each unobserved factor $f_j \notin \mathcal{E}$, $\sum_{f_j \in \{0,1\}} P(f_j \mid L_{\pi(j)}) = 1$, all such terms vanish from the product, leaving only factors in $\mathcal{E}$. Grouping the remaining product by latent index $i$ yields

$$P(\mathcal{E} \mid O_r) = \sum_{\ell \in \{0,1\}^k} \prod_{i=1}^{k} \left( P(L_i = \ell_i \mid O_r) \prod_{f \in \mathcal{E}_i} P(f = 1 \mid L_i = \ell_i) \right).$$

Finally, the summation over $\ell$ factorizes into $k$ independent binary sums (one per $L_i$), giving (4). Equation (5) follows by Bayes' rule and the uniform prior over outcomes. $\qquad \square$

A.2.3. JUSTIFYING THE PROXY-LIKELIHOOD MAPPING AND ITS SENSITIVITY.

Eliciting $\phi_f = P(O_1 \mid f = 1)$ does not, by itself, uniquely identify the likelihoods $P(f = 1 \mid O_k)$ without additional assumptions. We adopt a symmetric evidence-channel assumption that is natural when a factor is interpreted as directional support for $O_1$ versus $O_2$:

$$P(f = 1 \mid O_1) = \theta_f, \qquad P(f = 1 \mid O_2) = 1 - \theta_f, \tag{6}$$

together with a uniform prior $P(O_1) = P(O_2) = 1/2$. Under (6), Bayes' rule yields

$$P(O_1 \mid f = 1) = \frac{\theta_f \cdot \frac{1}{2}}{\theta_f \cdot \frac{1}{2} + (1 - \theta_f) \cdot \frac{1}{2}} = \theta_f,$$

hence $\theta_f = \phi_f$ exactly. Equivalently, with a uniform prior, the elicited posterior implies a Bayes factor (likelihood ratio)

$$\mathrm{LR}_f \triangleq \frac{P(f = 1 \mid O_1)}{P(f = 1 \mid O_2)} = \frac{P(O_1 \mid f = 1)}{P(O_2 \mid f = 1)} = \frac{\phi_f}{1 - \phi_f},$$

and (6) selects the unique likelihood pair consistent with this $\mathrm{LR}_f$ while preserving symmetry.

When the symmetry assumption is violated, the impact of this proxy can be analyzed in log-odds space. Let $\mathrm{logit}(p) \triangleq \log \frac{p}{1-p}$. For NB with positive evidence $\mathcal{E}$, the posterior log-odds decomposes additively:

$$\mathrm{logit}\, P(O_1 \mid \mathcal{E}) = \mathrm{logit}\, P(O_1) + \sum_{f \in \mathcal{E}} \log \mathrm{LR}_f.$$

Therefore, if our implied $\widehat{\mathrm{LR}}_f = \phi_f/(1 - \phi_f)$ differs from the true $\mathrm{LR}_f^\star$, the induced log-odds error is bounded by

$$\left| \mathrm{logit}\, \widehat{P}(O_1 \mid \mathcal{E}) - \mathrm{logit}\, P^\star(O_1 \mid \mathcal{E}) \right| \le \sum_{f \in \mathcal{E}} \left| \log \widehat{\mathrm{LR}}_f - \log \mathrm{LR}_f^\star \right|.$$

This highlights a failure mode where a few mis-elicited, high-weight factors dominate the posterior. In practice, we mitigate this via probability clipping/smoothing, robust factor filtering, and aggregation with the CBN through LOP, which reduces reliance on any single factor's likelihood proxy.

A.2.4. OUTCOME CPT DERIVED FROM $P(L_i \mid O_k)$ VIA BAYES' RULE

In the main text, we also describe parameterizing the CBN by assigning priors to latent nodes and deriving $P(\text{Outcome} \mid L_1, \dots, L_k)$ from $P(L_i \mid O_k)$ via Bayes' rule. The following lemma formalizes the conversion under the conditional independence implied by (3).

**Lemma A.2** (Bayes-rule derivation of $P(\text{Outcome} \mid \mathcal{L})$). *Assume $P(\text{Outcome} = O_1) = P(\text{Outcome} = O_2) = \frac{1}{2}$ and that $L_1, \dots, L_k$ are conditionally independent given Outcome, i.e., $P(\mathcal{L} \mid O_k) = \prod_{i=1}^{k} P(L_i \mid O_k)$. Then for any latent assignment $\ell \in \{0, 1\}^k$,*

$$P(\text{Outcome} = O_1 \mid \mathcal{L} = \ell) = \frac{\prod_{i=1}^{k} P(L_i = \ell_i \mid O_1)}{\prod_{i=1}^{k} P(L_i = \ell_i \mid O_1) + \prod_{i=1}^{k} P(L_i = \ell_i \mid O_2)}. \tag{7}$$

**Proof of Lemma A.2.** By Bayes' rule,

$$P(\text{Outcome} = O_1 \mid \mathcal{L} = \ell) = \frac{P(\mathcal{L} = \ell \mid O_1) P(O_1)}{P(\mathcal{L} = \ell \mid O_1) P(O_1) + P(\mathcal{L} = \ell \mid O_2) P(O_2)}.$$

Using the uniform prior and the conditional independence $P(\mathcal{L} \mid O_k) = \prod_i P(L_i \mid O_k)$ yields (7). $\qquad \square$

A.2.5. HOW SHARED LATENTS INDUCE DEPENDENCE AND WHEN NB OVERCOUNTS EVIDENCE

We next quantify the dependence induced by a shared latent parent and show how this can deviate from the NB product form.

**Proposition A.3** (Dependence induced by a shared latent parent). *Fix an outcome $O_k$ and consider two factors $f_a, f_b$ that share the same latent parent L. Let $\beta := P(L = 1 \mid O_k)$ and denote $\theta_a := P(f_a = 1 \mid L = 1)$, $\theta_b := P(f_b = 1 \mid L = 1)$, with $P(f = 1 \mid L = 0) = 1 - \theta$ as in the main text. Then*

$$P(f_a = 1, f_b = 1 \mid O_k) = \beta\,\theta_a\theta_b + (1 - \beta)\,(1 - \theta_a)(1 - \theta_b), \tag{8}$$
$$P(f_a = 1 \mid O_k) = \beta\,\theta_a + (1 - \beta)\,(1 - \theta_a), \tag{9}$$
$$P(f_b = 1 \mid O_k) = \beta\,\theta_b + (1 - \beta)\,(1 - \theta_b). \tag{10}$$

*Moreover, the covariance of $(f_a, f_b)$ under $O_k$ equals*

$$\mathrm{Cov}(f_a, f_b \mid O_k) = \beta(1 - \beta)\,(2\theta_a - 1)(2\theta_b - 1). \tag{11}$$

*In particular, if $\theta_a > 1/2$ and $\theta_b > 1/2$ (or both $< 1/2$), then $\mathrm{Cov}(f_a, f_b \mid O_k) > 0$, so the joint probability $P(f_a = 1, f_b = 1 \mid O_k)$ is larger than the NB product $P(f_a = 1 \mid O_k)P(f_b = 1 \mid O_k)$.*

**Proof of Proposition A.3.** Equations (8)–(10) follow by marginalizing the latent state:

$$P(f_a = 1, f_b = 1 \mid O_k) = \sum_{\ell \in \{0,1\}} P(L = \ell \mid O_k)\, P(f_a = 1 \mid L = \ell)\, P(f_b = 1 \mid L = \ell),$$

and similarly for each marginal. For the covariance,

$$\mathrm{Cov}(f_a, f_b \mid O_k) = P(f_a = 1, f_b = 1 \mid O_k) - P(f_a = 1 \mid O_k)P(f_b = 1 \mid O_k).$$

Substituting (8)–(10) and simplifying yields (11). The sign claim follows since $\beta(1 - \beta) > 0$ whenever $\beta \in (0, 1)$. □

**Implication.** When multiple positively-informative factors (with $\theta_f > 1/2$) share a latent parent, they are positively correlated under a fixed outcome. Treating them as independent (as in NB) can thus mischaracterize the likelihood of observing their conjunction and lead to systematic overconfident posteriors.

# B. Appendix B

## B.1. Parameter Initialization

**Factor prior initialization** Factor labels ("supports $O_1$", "supports $O_2$", "neutral") are assigned via the self-consistency procedure described in our method-Iterative Abduction. We then replace the LLM-elicited posterior $\theta_{f_j} = P(O_1 \mid f_j)$ with

$$P_{\mathrm{init}}(O_1 \mid f_j) = \begin{cases} 0.75, & f_j \text{ labeled "supports } O_1\text{"}, \\ 0.50, & f_j \text{ labeled "neutral"}, \\ 0.25, & f_j \text{ labeled "supports } O_2\text{"}. \end{cases}$$

**Latent CPT computation** Each latent variable $L_k$ aggregates a subset of factors whose labels yield counts

$$\mathtt{counts}[L_k] = \{\, \text{Outcome1} : c_k^1, \quad \text{Outcome2} : c_k^2, \quad \text{Neutral} : c_k^N \,\}.$$

We apply Laplace smoothing ($\epsilon$) and split neutrals evenly:

$$\tilde{c}_k^1 = c_k^1 + \tfrac{1}{2} c_k^N + \epsilon, \quad \tilde{c}_k^2 = c_k^2 + \tfrac{1}{2} c_k^N + \epsilon,$$

then compute

$$P(L_k = 1 \mid O_1) = \frac{\tilde{c}_k^1}{\tilde{c}_k^1 + \tilde{c}_k^2}, \quad P(L_k = 1 \mid O_2) = \frac{\tilde{c}_k^2}{\tilde{c}_k^1 + \tilde{c}_k^2}.$$

Enumerating all $2^n$ latent-state vectors $x \in \{0, 1\}^n$, we form the likelihoods $\prod_k P(L_k = x_k \mid O_i)$, multiply by the prior $P(O_i) = 0.5$, and normalize:

$$P(x \mid O_i) = \frac{P(O_i) \prod_k P(L_k = x_k \mid O_i)}{\sum_{j=1}^2 P(O_j) \prod_k P(L_k = x_k \mid O_j)}.$$

**Probability product approximation**   We approximate the marginal likelihood of evidence $\mathcal{E}$ under model $M$ by

$$P(\mathcal{E} \mid M) \approx P(O_1 \mid \mathcal{E}, M) \cdot P(O_2 \mid \mathcal{E}, M).$$

## B.2.   Aggregation Methods

**Linear Opinion Pool (LOP)**   Given per-model posteriors $P(O_1 \mid \mathcal{E}, M)$, LOP aggregates them as

$$P_{\text{LOP}}(O_1 \mid \mathcal{E}) = \sum_{M \in \{\text{NB,CBN}\}} w_M \, P(O_1 \mid \mathcal{E}, M).$$

We use fixed weights tailored to the underlying LLM: for Qwen2.5-32B, $w_{\text{NB}} = 0.8$ and $w_{\text{CBN}} = 0.2$; for Qwen2.5-72B, $w_{\text{NB}} = w_{\text{CBN}} = 0.5$.

**Bayesian Model Averaging (BMA)**   Under BMA, the final posterior is a weighted sum of each model's predictions:

$$P_{\text{BMA}}(O_1 \mid \mathcal{E}) = \sum_{M \in \{\text{NB,CBN}\}} P(M \mid \mathcal{E}) \, P(O_1 \mid \mathcal{E}, M),$$

where the model weights are proportional to the prior times the model evidence:

$$P(M \mid \mathcal{E}) \propto P(M) \, P(\mathcal{E} \mid M), \quad P(M) = \tfrac{1}{2}.$$

In practice, we approximate the evidence by the product of the two outcome posteriors, $P(\mathcal{E} \mid M) \approx P(O_1 \mid \mathcal{E}, M) \cdot P(O_2 \mid \mathcal{E}, M)$, and normalize these to obtain $P(M \mid \mathcal{E})$. Finally, the aggregated probability is

$$P_{\text{BMA}}(O_1 \mid \mathcal{E}) = \sum_M w_M \, P(O_1 \mid \mathcal{E}, M).$$

This "product-of-posteriors" is only an approximation: the true evidence requires summing over all latent or factor assignments, $P(\mathcal{E} \mid M) = \sum_x P(\mathcal{E}, x \mid M)$, which becomes intractable as the factor space grows.

## B.3.   Experiment Dataset Description

**COMMON2SENSE**   Singh et al. (2021) introduces a multi-domain commonsense reasoning and planning benchmark comprising true/false natural language statements. We select only the comparative reasoning instances in which a smaller pre-trained model shows low confidence. Each statement is paired with its GPT-4–generated opposite, yielding two outcomes per scenario. For each outcome, 10 supporting conditions are generated and then filtered via reverse verification to ensure quality. The original test split contained 350 instances; to enable a fairer evaluation, the first three authors of this paper manually expanded it to 530 instances—strictly by adding contrasting conditions to the existing scenarios and outcomes, following the annotation protocol of Feng et al. (2025). The final dataset comprises 216 scenarios and 3,822 instances, with an average of 9 conditions per outcome.

**TODAY**   Feng et al. (2023) proposes a temporal reasoning dataset where the effect of appending an extra sentence on temporal relations is evaluated. It consists of 1,000 instances designed to probe fine-grained temporal inference under controlled scenario modifications.

**PLASMA**   Brahman et al. (2023) focuses on plan revision under a new condition. For each of 279 scenarios, GPT-4 identifies and rewrites the altered step into two alternative outcomes. The less common outcome in each pair is selected, and 5 supporting conditions are generated to favor it. The resulting dataset comprises 279 scenarios and 1,395 instances, with exactly 5 conditions per outcome.

**Fact-Checking Benchmarks**   In addition, we sample 100 instances (with random seed fixed to 42) from the test split of each of four widely used fact-checking datasets, treating the document as the condition $U$ and the claim as the outcome $O$, with the central scenario $S_{\text{cen}}$ left empty. The long-document contexts pose a challenge for robust factor-space construction.

- **COVID** (Saakyan et al., 2021): Scientific claims about COVID-19 paired with biomedical abstracts, reflecting high-precision domain reporting.

- **ExpertQA** (Malaviya et al., 2024): Expert-driven statements with detailed contextual explanations spanning medicine, law, and engineering.

- **CNN** (Tang et al., 2024): News-derived claims with multi-sentence evidential contexts, suitable for long-range factual consistency evaluation.

- **XSum** (Tang et al., 2024): Headline-style summaries paired with full article text, emphasizing abstractiveness under concise outcomes.

Table 6 reports the average character lengths of outcome and condition in each dataset.

*Table 6.* Average character lengths of outcome and condition across all datasets.

| Dataset | Outcome | Condition |
|---|---|---|
| Common2Sense | 79.78 | 80.98 |
| Plasma | 92.69 | 79.76 |
| Today | 79.08 | 71.07 |
| COVID | 89.37 | 505.04 |
| ExpertQA | 163.20 | 2399.16 |
| CNN | 296.31 | 2972.89 |
| XSum | 127.25 | 1875.37 |

### B.4. Hyperparameter Settings

Table 7 provides a comprehensive list of the key hyperparameters used in our experiments. **Decision Stage.** For downstream fact-checking, we convert probabilistic outputs into categorical labels by applying a fixed decision threshold: a claim is marked as `support` only when its predicted probability exceeds $0.9$, otherwise it is treated as `unsupport`. Similar threshold-based rules are routinely adopted in recent fact-verification systems to balance precision and recall; for instance, AMREx employs dataset-specific entailment thresholds to separate "Supports", "Refutes", and "Not Enough Info" classes (Jayaweera et al., 2024).

### B.5. Clustering and LLM calibration bias

#### B.5.1. CLUSTERING QUALITY AND ALGORITHM CHOICE

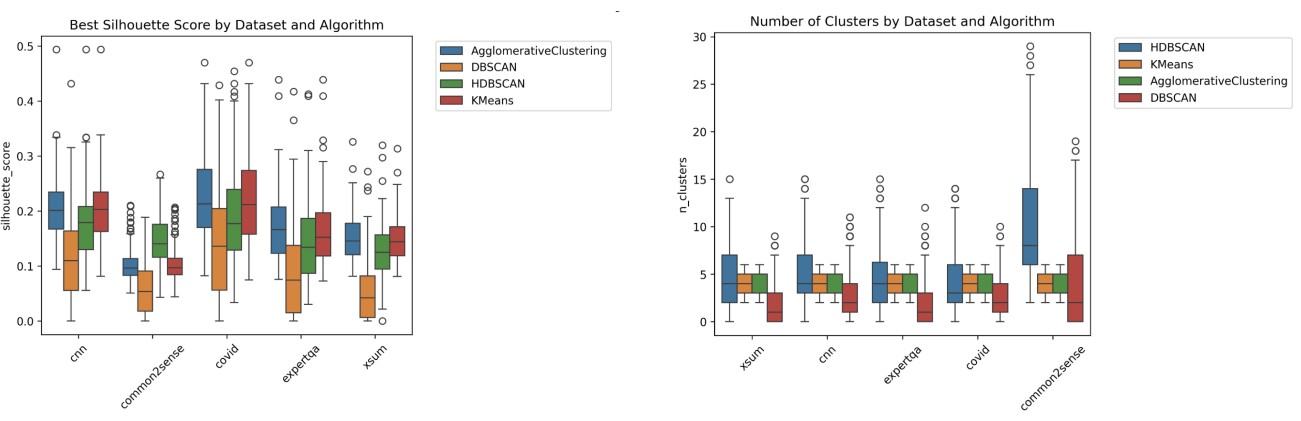

*(a)* Silhouette Score by Dataset and Algorithm    *(b)* Number of Clusters by Dataset and Algorithm

*Figure 4.* Comparison of clustering quality and flexibility across algorithms.

*Table 7.* Hyperparameter settings used in our experiments. Note that settings for factor generation vary by dataset group to accommodate different complexity levels, applies a decision threshold for support classification. Here $n_{\text{factors}}$ denotes the number of factors.

| Stage | Parameter | Value | Description |
|---|---|---|---|
| **Iterative Factor Generation** | | | |
| Factor Generation | *For Common2Sense & Plasma datasets:* | | |
| | Target factor count ($N_{target}$) | 80 | Target number of unique factors. |
| | Batch size ($b$) | 10 | Sentences generated per LLM call. |
| | Max rounds ($T_{\max}$) | 20 | Maximum number of generation rounds. |
| | *For Today, COVID, XSum, CNN, & ExpertQA datasets:* | | |
| | Target factor count ($N_{target}$) | 40 | Target number of unique factors. |
| | Batch size ($b$) | 5 | Sentences generated per LLM call. |
| | Max rounds ($T_{\max}$) | 10 | Maximum number of generation rounds. |
| **Context-Aware Factor Mapping** | | | |
| Hierarchical Retrieval | Top-$K_1$ clusters | 3 | Number of top clusters to retrieve in the coarse search. |
| | Top-$K_2$ factors | 5 | Number of top factors to retrieve per cluster in the fine search. |
| | Prototype weight ($\alpha$) | 0.5 | Weight balancing cluster theme vs. factor content in prototypes. |
| Self-Consistent Filtering | Voting rounds ($R$) | 3 | Number of LLM calls for the majority voting mechanism. |
| | Vote ratio | 0.5 | Ratio for calculating the vote threshold $\tau$. |
| **Probabilistic Inference** | | | |
| Parameter Elicitation | Smoothing alpha ($\epsilon$) | 0.5 | The Laplace smoothing factor for CPT stabilization. |
| | LLM sampling temp. | 0.5 | Sampling temperature for LLM-based parameter elicitation. |
| | LLM parse retries | 20 | Maximum retries for eliciting valid parameters from the LLM. |
| **Fact Checking (*COVID, XSum, CNN, & ExpertQA datasets*) threshold** | | | |
| Decision-making | Probability threshold ($\tau_{\text{dec}}$) | 0.9 | Minimum probability required to label as "support." |
| **Factor Embedding and Clustering (UMAP + HDBSCAN)** | | | |
| Embedding | UMAP $n_{\text{components}}$ | $\min(50, \max(10, \lfloor n_{\text{factors}}/5 \rfloor))$ | Embedding dim. |
| | UMAP $n_{\text{neighbors}}$ | $\min(15, n_{\text{factors}} - 1)$ | Local–global trade-off. |
| | UMAP metric | `cosine` | Neighbor distance. |
| Clustering | HDBSCAN min_cluster_size | $\max(2, \lfloor n_{\text{factors}}/20 \rfloor)$ | Min cluster size (else noise). |
| | HDBSCAN metric | `euclidean` | Clustering distance. |

We adopt **HDBSCAN** as our clustering algorithm due to its ability to automatically determine the number of clusters based on data density, which is especially important in open-ended and semantically diverse factor spaces. As shown in Figure 4(b), HDBSCAN produces a wider and more adaptive range of cluster counts across datasets compared to other algorithms.

Although fixed-$k$ algorithms like KMeans or Agglomerative Clustering occasionally yield higher silhouette scores on certain datasets, their performance heavily depends on the manually chosen number of clusters, which may not generalize across tasks. This fixed structure risks oversimplifying or overfragmenting the semantic space when the true number of factor types varies with input complexity.

DBSCAN also determines clusters automatically, but tends to merge semantically distinct groups or produce very few clusters, especially under high noise or sparse factor conditions.

Overall, HDBSCAN provides a good trade-off between clustering quality and adaptability, making it suitable for our pipeline that operates across datasets with diverse semantic granularity.

### B.5.2. EFFECT OF PROBABILITY-ESTIMATOR LLM AND CALIBRATION

To test how sensitive ANCHOR is to such calibration effects, we replace Qwen2.5-72B with the smaller GPT-4o-mini while keeping the factor space, mapping pipeline, and NB+CBN inference fixed, we evaluate on CNN, Today, Plasma, and ExpertQA; as shown in Table 8, all changes are within about ±2.5 points, indicating limited sensitivity to the specific choice of probability-estimator LLM.

We further analyze how changing the probability-estimator LLM reshapes factor-level scores: Table 9 reports distributional distances between factor-score distributions under Qwen2.5-72B and GPT-4o-mini, and Figure 5 visualizes the corresponding smoothed factor-level probability profiles on CNN, TODAY, PLASMA, and EXPERTQA, showing that only CNN exhibits a pronounced distributional shift while overall factor patterns remain similar, consistent with the small performance differences

*Table 8.* Task performance (%) of ANCHOR with different probability-estimator LLMs.

| Dataset | Qwen2.5-72B | GPT-4o-mini | $\Delta$ (mini $-$ 72B) |
|---|---|---|---|
| CNN | 62.9 | 62.3 | $-0.6$ |
| ExpertQA | 60.5 | 58.8 | $-1.7$ |
| Plasma | 76.2 | 73.8 | $-2.4$ |
| Today | 82.7 | 83.8 | $+1.1$ |

in Table 8.

*Table 9.* Distributional distances between factor scores under Qwen2.5-72B and GPT-4o-mini.

| Dataset | JS div. | Wass. dist. | KS $p$-value | Diff. |
|---|---|---|---|---|
| CNN | 0.356 | 0.074 | 0.0001 | significant |
| TODAY | 0.235 | 0.042 | 0.7012 | not significant |
| PLASMA | 0.205 | 0.040 | 0.7287 | not significant |
| EXPERTQA | 0.313 | 0.045 | 0.0773 | not significant |

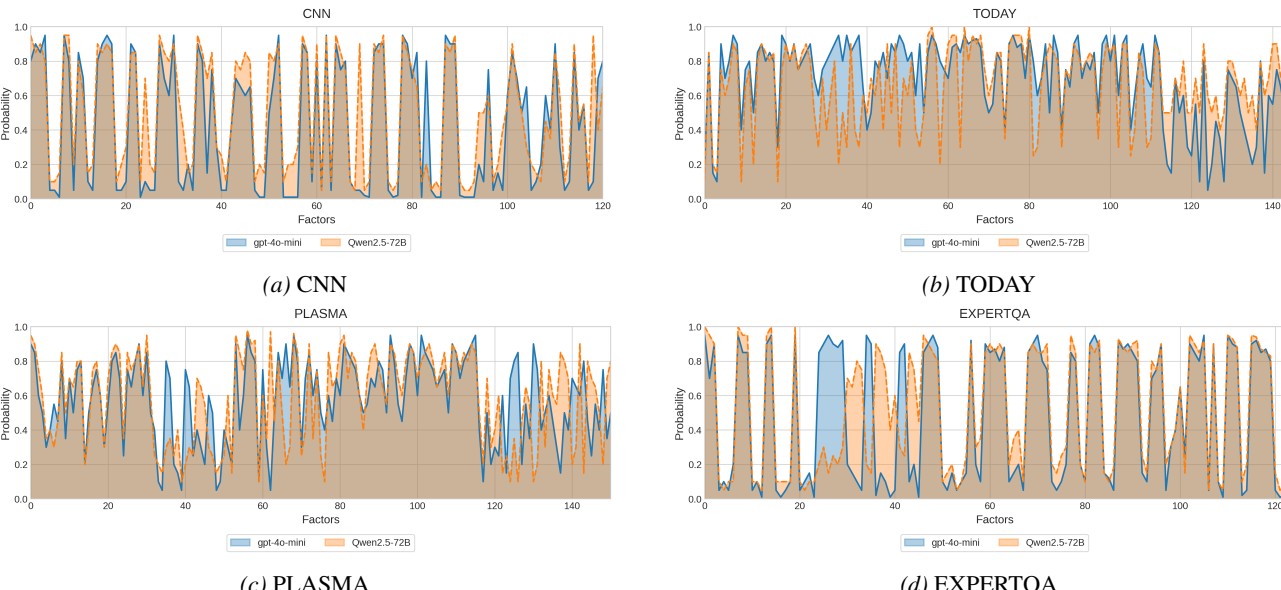

*Figure 5.* Smoothed factor-level probability profiles under Qwen2.5-72B and GPT-4o-mini on four datasets.

## B.6. Sensitivity to $K$ and Method Comparison

**Why we finally choose KNN and $(K_1{=}3,\ K_2{=}5)$** All analyses here are conducted on the ANCHOR model built on Qwen2.5-72B. KNN consistently delivers a low Unknown Rate and stable average F1 in our tests, avoiding BM25's high unknown proportion and FAISS's fluctuations. Very small $K$ values (2/3) under-cover relevant factors, while enlarging them to 4/8 actually degrades performance relative to our tuned $(3, 5)$ setting because more noisy neighbors are included. Hence, the mid-range $(3, 5)$ strikes a better coverage–precision trade-off. Moreover, KNN's Euclidean distance is robust to our factor-embedding scale and requires minimal tuning and engineering overhead. **Accordingly, the main results reported in Table 1 are obtained with KNN using $(K_1{=}3,\ K_2{=}5)$.**

## B.7. Additional Details for the BIRD-on-ANCHOR Ablations

**Complete-information spaces in BIRD vs. ANCHOR.** Conceptually, BIRD assumes that each factor takes two mutually exclusive attribute values. For a given scenario, these categorical factors form a low-arity product space, so the complete-information training set can be obtained by (conceptually) enumerating this Cartesian product and sampling from it. In

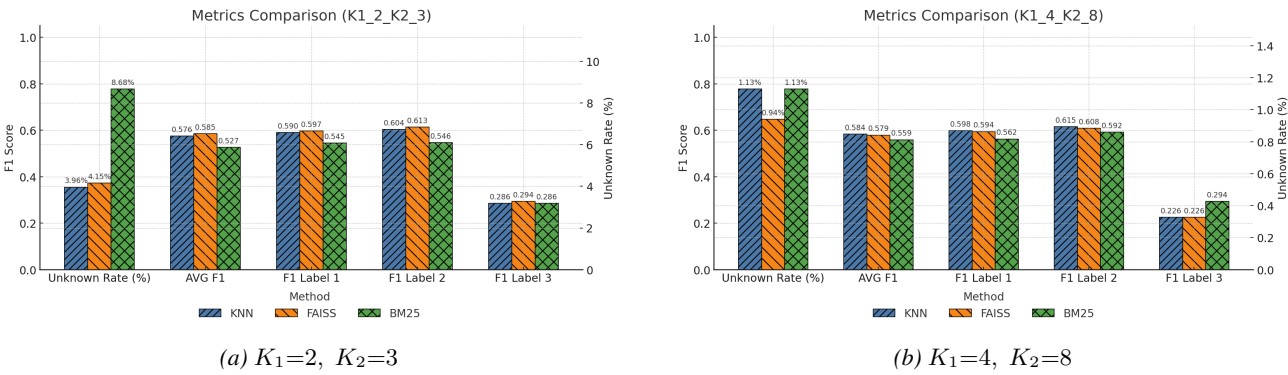

*Figure 6.* Unknown Rate and per-class F1 comparison across KNN, FAISS, and BM25 under two $K$ settings.

ANCHOR, factors are standalone evidence-like statements drawn from a global pool and organized into clusters and themes, not tied to a fixed two-value choice per slot. A complete-information assignment is thus a subset of this pool, and naively forming a Cartesian product over clusters would both generate many incoherent combinations and lead to a combinatorial explosion.

**Structured Monte Carlo scheme for CPT training.**    To plug BIRD's CPT-based outcome model into ANCHOR's factor space, we therefore adopt a structured Monte Carlo scheme. For each mapped condition, we fix a "base" set of factors returned by the mapping pipeline, randomly complete it with a bounded number of additional factors up to a maximum length $L$, and treat the resulting complete assignments $f^{(1)}, \ldots, f^{(m)}$ as samples from a distribution $\pi$ over the intractable complete-information space. This allows us to approximate the BIRD training objective

$$\mathcal{L}(\theta) = \mathbb{E}_{f \sim \pi}\big[\ell(P_\theta(O \mid f), y(f))\big] \approx \hat{\mathcal{L}}_m(\theta) = \frac{1}{m} \sum_{i=1}^{m} \ell\big(P_\theta(O \mid f^{(i)}), y(f^{(i)})\big),$$

where $P_\theta(O \mid f)$ is the outcome model with parameters $\theta$, $\ell$ is the training loss, and $y(f^{(i)})$ is the LLM's coarse probability assessment under complete information $f^{(i)}$. In the experiments we match the original BIRD setting with $m = 128$ sampled assignments per scenario.

**Motivation for LLM-based priors.**    This Monte Carlo construction is practical but imperfect: with a finite sampling budget, some factors may never appear in the training set and remain tied to their priors, and uniform completion from a large factor pool can under-cover parts of the space or induce slightly unrealistic co-occurrences. Together with prior work showing that LLMs can provide informative priors and probabilistic knowledge for Bayesian models (Riegler et al., 2025; Nafar et al., 2025; Capstick et al., 2025), this motivates ANCHOR's main design choice of eliciting coarse factor-level probabilities directly from the LLM instead of relying purely on CPT fitting.

### B.8. Limitations and Future Work

**Limitations.**    Our dataset relies on scenario-local pairwise comparisons, where opposing statements are compared only within the same scenario. This creates many small and weakly connected subgraphs, limiting the usefulness of global ranking methods such as Bradley–Terry and weakening ECE-style calibration analysis due to sparse, non-comparable confidence bins. Moreover, although LLMs are used only for coarse CPT initialization, the resulting estimates may still suffer from miscalibration and prompt sensitivity. Future work can consider denser globally comparable comparisons and more robust probability acquisition strategies, such as empirical-frequency anchoring, held-out calibration, or alternative elicitation objectives.

**Toward adaptive and auditable reasoning.**    Beyond probability elicitation, future work can make the inference pipeline more adaptive, verifiable, and auditable. Recent studies offer complementary directions, including causal reflection and verification (Lin et al., 2026b), external semantic auditing (Xu et al., 2026), multi-path reasoning aggregation (Luo et al., 2026a), programmatic synthesis with consistency verification (Liu et al., 2026), token-aware optimization (Liu et al., 2025), self-evolution trajectory optimization (Lin et al., 2025), curriculum-based alignment with AI feedback (Lin et al.,

2026a). These ideas could strengthen our framework through more reliable factor–context mapping, CPT initialization, trajectory-level verification, Bayesian aggregation, adaptive supervision, and efficient deployment.

## C. Appendix C

### C.1.  Comprehensive Evaluation of Factor Spaces and Mappings

In this subsection, we first compare the factor spaces generated by ANCHOR and BIRD (Table10), then examine their mapping performance under specific cooking conditions (Table11), and finally analyze the organization of latent nodes for Outcome1 (Table12).

*Table 10.* An example of ANCHOR and BIRD generated factor spaces

| | |
|---|---|
| **Scenario**: The efficiency of cooking noodles is being compared between hot water and warm water. **Outcome1**: Noodles cook much more efficiently in hot water than they do in warm water. **Outcome2**: Noodles cook much more efficiently in warm water than they do in hot water. | |
| **ANCHOR Generated Factor Space** | |
| **Clumping Prevention** | hard inside; reduced sticking; clumping reduction; rapid softening |
| **Efficiency Reductions** | energy and time efficiency; practicality; energy efficiency |
| **Texture & Flavor** | texture preservation; al dente texture; flavor; dispersing flavorings; texture inconsistency; uniform texture; longer gelatinization; palatability; better flavor infusion |
| **Cooking Efficiency** | ensures even cooking and texture retention; inefficient activation of cooking process; professional kitchens; cooking environment; slow outer layer cooking; reduced cooking time; gradual cooking; difficulty controlling cooking process |
| **Starch Digestion Efficiency** | effective breakdown of starches; starch breakdown inefficiency; digestibility; gluten breakdown |
| **Temperature Control** | inconsistent temperature control; insufficient thermal energy; lower temperature; consistent use of hot water; noodle hydration; evaporation; temperature of water; pre-soaking benefits; delicate noodles |
| **Cooking Safety & Precision** | risk of undercooking or overcooking; avoiding overcooking/mushiness; food safety (reduces risk of foodborne illness); overcooking prevention |
| **BIRD Generated Factor Space** | |
| **Cooking Temperature and Time** | Hot water significantly reduces cooking time by breaking down starches more quickly; Warm water requires a longer cooking time for noodles to become tender |
| **Noodle Hydration Process** | Hot water leads to faster hydration and cooking of noodles; Warm water results in a slower but more even hydration process |
| **Recommended Cooking Time** | Shorter recommended cooking time in hot water; Longer recommended cooking time in warm water |
| **Energy Efficiency** | Hot water uses more energy to heat up initially but cooks faster, potentially saving overall energy; Warm water uses less energy to heat up but requires a longer cooking time, potentially using more energy overall |
| **Texture and Quality** | Hot water can lead to a more uniform texture and prevent clumping of noodles; Warm water may result in a softer texture but can cause the noodles to become mushy if cooked for too long |
| **Cooking Efficiency** | Significantly reduces cooking time with hot water; Requires longer cooking time with warm water |
| **Noodle Texture** | Hot water ensures a more uniform texture; Warm water can lead to a softer texture |
| **Starch Breakdown** | High temperature in hot water breaks down starches more quickly; Lower temperature in warm water results in slower starch breakdown |
| **Energy Consumption** | Hot water uses more initial energy but cooks faster; Warm water uses less initial energy but cooks longer |
| **Hydration Process** | Hot water leads to faster hydration; Warm water results in a slower but more even hydration |
| **Clumping Prevention** | Hot water helps prevent noodles from clumping; Warm water may increase the risk of noodles clumping |

*Table 11.* An example of BIRD and ANCHOR conditions factor mappings

**Scenario**: The efficiency of cooking noodles is being compared between hot water and warm water.

**Outcome1**: Noodles cook much more efficiently in hot water than they do in warm water.

**Outcome2**: Noodles cook much more efficiently in warm water than they do in hot water.

| Condition | BIRD Mapped Factors | ANCHOR Mapped Factors |
|---|---|---|
| **The higher temperature of hot water can accelerate the gelatinization process in starchy foods like noodles.** | Shorter recommended cooking time in hot water; High temperature in hot water breaks down starches more quickly; Hot water can lead to a more uniform texture and prevent clumping of noodles; Hot water leads to faster hydration and cooking of noodles | noodle hydration; temperature of water; effective breakdown of starches; digestibility; better flavor infusion; dispersing flavorings; flavor |
| **Cooking in hot water reduces the amount of total time that the noodles spend in the water.** | Shorter recommended cooking time in hot water; High temperature in hot water breaks down starches more quickly; Significantly reduces cooking time with hot water; Hot water leads to faster hydration and cooking of noodles; Hot water leads to faster hydration; Hot water significantly reduces cooking time by breaking down starches more quickly | reduced cooking time; ensures even cooking and texture retention; noodle hydration; temperature of water; avoiding overcooking/mushiness; overcooking prevention |
| **Hot water helps in killing any potential foodborne pathogens or microbes present in the noodles.** | None | consistent use of hot water; temperature of water; food safety (reduces risk of foodborne illness) |
| **You're cooking a large quantity of noodles.** | None | gradual cooking; inefficient activation of cooking process; ensures even cooking and texture retention; cooking environment; avoiding overcooking/mushiness; overcooking prevention; risk of undercooking or overcooking; starch breakdown inefficiency; effective breakdown of starches |
| **You are at a high altitude where water boils at lower temperatures than at sea level.** | None | temperature of water; lower temperature; insufficient thermal energy; risk of undercooking or overcooking; slow outer layer cooking; cooking environment; inefficient activation of cooking process; consistent use of hot water; evaporation; overcooking prevention; avoiding overcooking/mushiness; gradual cooking |

As shown in Table 10, ANCHOR produces a more hierarchically structured set of categories that cover safety and nutrition dimensions while maintaining low redundancy, whereas BIRD's space exhibits multiple synonymous and fragmented entries. Table 11 demonstrates that ANCHOR successfully maps key factors—such as food safety, temperature control, and

process precision—under conditions like pathogen elimination at high temperatures, large-batch cooking, and high-altitude environments, filling gaps left by BIRD. Furthermore, Table 12 presents ANCHOR's five latent dimensions for Outcome1 (TextureLat, FlavorLat, EfficiencyLat, SafetyLat, and ProcessControlLat), each with a logically coherent and balanced distribution of factors, offering robust support for downstream factor analysis and visualization. Overall, the ANCHOR method shows clear advantages in generating a high-cohesion, low-redundancy, and practically actionable factor space.

*Table 12.* An example of latent nodes organized under parent Outcome1 with their corresponding factors

**Scenario**: The efficiency of cooking noodles is being compared between hot water and warm water.

**Outcome1**: Noodles cook much more efficiently in hot water than they do in warm water.

**Outcome2**: Noodles cook much more efficiently in warm water than they do in hot water.

| Parent Node | Latent Node | Child Nodes (Factors) |
|---|---|---|
| Outcome1 | TextureLat | slow outer layer cooking; texture preservation; ensures even cooking and texture retention; al dente texture; reduced cooking time; overcooking prevention; uniform texture; gradual cooking; palatability; hard inside; starch breakdown inefficiency; reduced sticking; rapid softening; delicate noodles; avoiding overcooking/mushiness; gluten breakdown; longer gelatinization; texture inconsistency |
| Outcome1 | FlavorLat | noodle hydration; flavor; dispersing flavorings; better flavor infusion; digestibility; food safety (reduces risk of foodborne illness) |
| Outcome1 | EfficiencyLat | energy and time efficiency; energy efficiency; evaporation; practicality; pre-soaking benefits; consistent use of hot water; cooking environment |
| Outcome1 | SafetyLat | temperature of water; lower temperature; inconsistent temperature control; insufficient thermal energy |
| Outcome1 | ProcessControlLat | clumping reduction; effective breakdown of starches; inefficient activation of cooking process; risk of undercooking or overcooking; difficulty controlling cooking process; professional kitchens |

## C.2. Detailed Algorithm Specifications

This section provides the detailed pseudo-code for the three core stages of the ANCHOR framework, as described in Section 4. Algorithm 1 details the construction of the hierarchical factor space. It operates by iteratively generating raw factors, validating them via self-consistency voting, and finally organizing them using a pipeline of embedding, clustering, and LLM-based thematic pruning. Algorithm 2 specifies the multi-stage process for mapping a condition to a relevant factor set. The pipeline begins with a broad, two-level hierarchical retrieval to generate candidates, followed by a voting-based filtering step and a final reflective refinement to ensure high precision. Algorithm 3 outlines the procedure for transforming a set of mapped factors into a final probability. It involves parameterizing two parallel models—Naïve Bayes and a latent-augmented Causal Bayesian Network—with probabilities elicited from an LLM, and then aggregating their posteriors for a robust final estimate.

## C.3. Prompts

## C.4. LLM Usage Statement

We clarify the role of LLMs in our research workflow. Specifically, LLMs were only used to assist with (i) code writing (e.g., generating boilerplate code, debugging minor errors, and improving readability), and (ii) polishing the writing of the paper text (e.g., improving clarity, grammar, and style). No experimental results, theoretical analyses, or substantive scientific claims in this paper were produced by LLMs. All methodological designs, experiments, and conclusions are solely the work of the authors.

---

**Algorithm 1** Iterative Factor Generation via Bottom-up Abduction

---

1: **Input:** Scenario description $s$, primary outcome $O_1$, secondary outcome $O_2$, LLM $\mathcal{M}$,
2:      target factor count $N_{target}$, batch size $b$, maximum rounds $T_{\max}$, clustering flag $C$
3: **Output:** Hierarchical factor space $\widetilde{\mathcal{F}}$, factor–outcome mapping $M$, clustering statistics
4:
5:  $\mathcal{F}^{(0)} \leftarrow \emptyset$
6: **for** $t \leftarrow 1$ **to** $T_{\max}$ **do**
7:    **if** $|\mathcal{F}^{(t-1)}| \geq K$ **then**
8:       **break**
9:    **end if**
10:    *// Contextual sentence generation*
11:    $S^{(t)} \leftarrow \text{GENERATESENTENCES}(s, O_1, O_2, b, \mathcal{M})$
12:    *// Factor harvesting and validation*
13:    $\Delta\mathcal{F}^{(t)} \leftarrow \text{EXTRACTFACTORS}(S^{(t)}, \mathcal{M})$
14:    $\mathcal{F}^{(t)} \leftarrow \mathcal{F}^{(t-1)} \cup \Delta\mathcal{F}^{(t)}$
15: **end for**
16:
17: $M \leftarrow \text{VOTESUPPORT}(\mathcal{F}^{(T)}, s, O_1, O_2, \mathcal{M})$
18: $\mathcal{F}_{\text{validated}} \leftarrow \{f \in \mathcal{F}^{(T)} \mid M[f] \in \{\text{``}O_1\text{''}, \text{``}O_2\text{''}, \text{``Neutral''}\}\}$
19:
20: **if** $C = \text{True}$ **then**
21:    *// Encode factors into dense representations*
22:    $E \leftarrow \{e_f \mid f \in \mathcal{F}_{\text{validated}}\}$ **using** MiniLM
23:    *// Dimensionality reduction and clustering*
24:    $E_{\text{reduced}} \leftarrow \text{UMAP}(E)$
25:    $\mathcal{C}_{\text{raw}} \leftarrow \text{HDBSCAN}(E_{\text{reduced}})$
26:    *// LLM-guided thematic organization and pruning*
27:    $\mathcal{C}_{\text{themed}} \leftarrow \text{ASSIGNTHEMES}(\mathcal{C}_{\text{raw}}, \mathcal{M})$
28:    $\widetilde{\mathcal{F}} \leftarrow \text{PRUNEREDUNDANCY}(\mathcal{C}_{\text{themed}}, M, \mathcal{M})$
29:    compute clustering statistics
30: **else**
31:    $\widetilde{\mathcal{F}} \leftarrow \{(\text{``default''}, \mathcal{F}_{\text{validated}})\}$
32:    set default clustering statistics
33: **end if**
34:
35: **return** $(\widetilde{\mathcal{F}}, M, \text{clustering statistics})$

---

---

**Algorithm 2** Context-Aware Factor Mapping

---

1: **Input:** Hierarchical factor space $\widetilde{\mathcal{F}} = \{(C_j, F_j)\}$, unclustered factors $F_u$, condition $u$,
2:      cluster top-$K_1$, factor top-$K_2$, voting rounds $R$, vote threshold $\tau$, balance weight $\alpha$
3: **Output:** High-precision factor set $\mathcal{F}^*(u)$
4:
5: $\mathcal{F}_{\text{cand}} \leftarrow \emptyset$
6:
7: **if** $\widetilde{\mathcal{F}} \neq \emptyset$ **then**
8:      *// Compute cluster prototypes*
9:      **for all** $(C_j, F_j) \in \widetilde{\mathcal{F}}$ **do**
10:        $e_{C_j} \leftarrow \alpha \cdot \text{EMBED}(\text{theme}(C_j)) + (1 - \alpha) \cdot \frac{1}{|F_j|} \sum_{f \in F_j} \text{EMBED}(f)$
11:      **end for**
12:      *// Build KNN indices*
13:      $\text{KNN}_{\text{clusters}} \leftarrow \text{BUILDKNN}(\{e_{C_j}\}, K_1)$
14:      **for all** $(C_j, F_j) \in \widetilde{\mathcal{F}}$ **do**
15:        $\text{KNN}_{F_j} \leftarrow \text{BUILDKNN}(\{\text{EMBED}(f) \mid f \in F_j\}, K_2)$
16:      **end for**
17:      *// Coarse-to-fine retrieval*
18:      $C_{\text{sel}} \leftarrow \text{KNN}_{\text{clusters}}.\text{QUERY}(\text{EMBED}(u))$
19:      **for all** $C_j \in C_{\text{sel}}$ **do**
20:        $F_{\text{retrieved}} \leftarrow \text{KNN}_{F_j}.\text{QUERY}(\text{EMBED}(u))$
21:        $\mathcal{F}_{\text{cand}} \leftarrow \mathcal{F}_{\text{cand}} \cup F_{\text{retrieved}}$
22:      **end for**
23: **end if**
24:
25: **if** $F_u \neq \emptyset$ **then**
26:      $\text{KNN}_{\text{unclustered}} \leftarrow \text{BUILDKNN}(\{\text{EMBED}(f) \mid f \in F_u\}, K_2)$
27:      $\mathcal{F}_{\text{cand}} \leftarrow \mathcal{F}_{\text{cand}} \cup \text{KNN}_{\text{unclustered}}.\text{QUERY}(\text{EMBED}(u))$
28: **end if**
29:
30: initialize vote counts $v[f] \leftarrow 0$ for all $f \in \mathcal{F}_{\text{cand}}$
31: **for** $r \leftarrow 1$ **to** $R$ **do**
32:      $S^{(r)} \leftarrow \text{LLMSELECT}(u, \mathcal{F}_{\text{cand}}, \mathcal{M})$
33:      **for all** $f \in S^{(r)}$ **do**
34:        $v[f] \leftarrow v[f] + 1$
35:      **end for**
36: **end for**
37: $\mathcal{F}_{\text{vote}} \leftarrow \{f \in \mathcal{F}_{\text{cand}} \mid v[f] \geq \tau\}$
38:
39: $\mathcal{F}^*(u) \leftarrow \text{REFLECTIVEREFINE}(u, \mathcal{F}_{\text{vote}}, \mathcal{M})$
40: **return** $\mathcal{F}^*(u)$

---

---

**Algorithm 3** Probabilistic Inference with Elicited Parameters

---

 1: **Input:** Mapped factor set $\mathcal{F}^*(u)$, outcome hypotheses $\{O_1, O_2\}$, LLM $\mathcal{M}$,
 2:      aggregation weights $w_{\text{NB}}, w_{\text{CBN}}$, smoothing parameter $\epsilon$
 3: **Output:** Aggregated posterior probability $P_{\text{agg}}(O_1 \mid \mathcal{F}^*(u))$
 4:
 5: *// Factor-level parameters for both models*
 6: **for all** $f \in \mathcal{F}^*(u)$ **do**
 7:    $\theta_f \leftarrow \text{ELICITPOSTERIOR}(f, O_1, \mathcal{M})$                                                  // $\theta_f \approx P(O_1 \mid f)$
 8:    $\theta_f \leftarrow \text{SMOOTH}(\theta_f, \epsilon)$
 9: **end for**
10:
11: *// Build NB model with conditional independence assumption*
12: Build NB structure: $O \rightarrow f$ for all $f \in \mathcal{F}^*(u)$
13: **for all** $f \in \mathcal{F}^*(u)$ **do**
14:    Set CPT: $P(f \mid O_1) = \theta_f$, $P(f \mid O_2) = 1 - \theta_f$
15: **end for**
16: Set uniform prior: $P(O_1) = P(O_2) = 0.5$
17: $P_{\text{NB}} \leftarrow \text{INFERPOSTERIOR}(\text{NB}, \mathcal{F}^*(u))$ using Eq. 2
18:
19: *// Latent variable discovery and structure learning*
20: $\mathcal{L} \leftarrow \text{ELICITLATENTVARIABLES}(\mathcal{F}^*(u), \mathcal{M})$
21: partition $\leftarrow \text{ASSIGNFACTORSTOLATENTS}(\mathcal{F}^*(u), \mathcal{L}, \mathcal{M})$
22:
23: *// Latent-level parameter elicitation*
24: **for all** $L_i \in \mathcal{L}$ **do**
25:    $p_{i1} \leftarrow \text{ELICITCONDITIONAL}(L_i, O_1, \mathcal{M})$                            // $p_{i1} = P(L_i = 1 \mid O_1)$
26:    $p_{i2} \leftarrow \text{ELICITCONDITIONAL}(L_i, O_2, \mathcal{M})$                            // $p_{i2} = P(L_i = 1 \mid O_2)$
27:    $(p_{i1}, p_{i2}) \leftarrow \text{SMOOTH}((p_{i1}, p_{i2}), \epsilon)$
28: **end for**
29:
30: *// Build CBN structure and parameterize CPTs*
31: Build CBN structure: $L_i \rightarrow f$ for $f \in \text{partition}(L_i)$, and $L_i \rightarrow O$ for all $L_i$
32: **for all** $L_i \in \mathcal{L}$ **do**
33:    Set uniform prior: $P(L_i = 1) = 0.5$
34:    **for all** $f \in \text{partition}(L_i)$ **do**
35:       Set CPT: $P(f \mid L_i = 1) = \theta_f$, $P(f \mid L_i = 0) = 1 - \theta_f$
36:    **end for**
37: **end for**
38: Derive outcome CPT: $P(O \mid L_1, \ldots, L_k)$ using $\{p_{i1}, p_{i2}\}_{i=1}^{k}$ via Bayes' rule
39: $P_{\text{CBN}} \leftarrow \text{INFERPOSTERIOR}(\text{CBN}, \mathcal{F}^*(u))$ using variable elimination
40:
41: $P_{\text{agg}} \leftarrow w_{\text{NB}} \cdot P_{\text{NB}} + w_{\text{CBN}} \cdot P_{\text{CBN}}$
42: **return** $P_{\text{agg}}$

---

**Example Prompt for Generating Supporting/Refuting Sentences**

**System**
You are an AI assistant that helps people make decisions.
Generate {n} diverse supporting or refuting sentences for scenario: {scenario},
comparing '{Outcome}' vs '{oppo_Outcome}'.

**User**
Scenario: Alice is training for a marathon.
Outcome: Running on a treadmill improves her endurance.
Opposite Outcome: Running on a treadmill does not improve her endurance.
Generate 2 sentences.

**Assistant**
1. Treadmill training allows Alice to maintain a consistent pace and monitor heart
rate, boosting her aerobic capacity.
2. The treadmill's adjustable incline simulates hill workouts, increasing leg strength
and stamina.

**User**
Scenario: Bob studies every evening.
Outcome: Studying in short, focused bursts enhances retention.
Opposite Outcome: Studying in short, focused bursts does not enhance retention.
Generate 2 sentences.

**Assistant**
1. Brief study sessions with breaks prevent mental fatigue and improve long-term
recall.
2. Frequent reviews in small intervals reinforce memory pathways, aiding retention.

*Figure 7.* Example Prompt for Generating Supporting or Refuting Sentences

**Few-Shot Examples for Factor Extraction**

**System**
Extract distinct factors from these sentences.  Think step by step about what factors
are mentioned, then provide your final answer as a JSON array.

**User**
Extract distinct factors from these sentences as a JSON array.
1.  Treadmill training allows maintaining a consistent pace and monitoring heart rate,
boosting aerobic capacity.
2.  The treadmill's adjustable incline simulates hill workouts, increasing leg strength
and stamina.

**Assistant**
Let me analyze these sentences to identify the key factors:
• Pace consistency (ability to maintain steady speed)
• Heart rate monitoring (tracking cardiovascular response)
• Adjustable incline (variable difficulty/terrain simulation)
• Leg strength (muscle development)
Final answer:  ["Pace consistency","Heart rate monitoring",
"Adjustable incline","Leg strength"]

**User**
Extract distinct factors from these sentences as a JSON array.
1.  Group work requires coordination between team members.
2.  Individual tasks allow for personal control and efficiency.

**Assistant**
Let me identify the key factors from these sentences:
• Coordination requirements (need for team synchronization)
• Personal control (individual autonomy)
• Efficiency (productivity/effectiveness)
• Team collaboration (working together)
Final answer:  ["Coordination requirements","Personal control",
"Efficiency","Team collaboration"]

*Figure 8.* Few-shot prompt–response pairs for factor extraction.

---

**Few-Shot Examples for Factor–Outcome Voting**

```
System
Decide which outcome the factor supports.  Reason briefly (1-2 sentences), then provide
your final answer as a JSON object.  Keep the explanation as short as possible--no
extra commentary.

User
Scenario:  Alice trains for a marathon.
Outcome1:  Treadmill running improves endurance.
Outcome2:  Treadmill running does not improve endurance.
Factor:  Pace consistency
Decide which outcome this factor supports:  Outcome1, Outcome2, or Both.  Think step by
step, then answer in JSON.

Assistant
Pace consistency forces a steady speed, building aerobic capacity and sustained effort.
Final answer:  {"Pace consistency": "Outcome1"}

User
Scenario:  Alice trains for a marathon.
Outcome1:  Treadmill running improves endurance.
Outcome2:  Treadmill running does not improve endurance.
Factor:  Weather conditions
Decide which outcome this factor supports:  Outcome1, Outcome2, or Both.  Think step by
step, then answer in JSON.

Assistant
Treadmill gives consistent conditions, yet outdoor weather readies race adaptability.
Final answer:  {"Weather conditions": "Both"}
```

*Figure 9.* Few-shot prompt–response pairs for factor–outcome voting.

---

**Few-Shot Examples for Theme Name Generation**

```
System
Generate a concise English theme name (1-3 words) that captures the common topic of
these factors.
Return only the theme name, no explanation.

User
Generate a theme name for these related factors:
["energy expenditure", "energy transfer efficiency"]

Assistant
Energy Efficiency

User
Generate a theme name for these related factors:
["precision control", "better control (accuracy)"]

Assistant
Control Precision
```

*Figure 10.* Few-shot prompt–response pairs for generating concise theme names.

**Few-Shot Examples for Factor–Condition Mapping**

**System**
You are an expert at analyzing logical relationships between conditions and factors. Given a condition and a list of candidate factors, select factors that have reasonable connections to the condition. Be inclusive rather than restrictive.

**User**
Scenario: A student is preparing for final exams.
Condition: The student spends more time in the library.
Candidate factors: ["Better time management", "More stress", "Increased social activities"]
Task: Select the factor list that is most reasonably related to the given condition. Think step by step about each factor's relevance, then provide your selection. Please respond with your reasoning followed by Final answer: and a JSON object in this exact format: {"answer": ["factor1", "factor2", ...]}.

**Assistant**
Let me analyze each factor: • "Better time management": Spending more time in the library suggests the student is organizing their study schedule better and dedicating focused time to studying. This is directly related. • "More stress": While exam preparation can be stressful, spending more time in the library is typically a positive study behavior, not necessarily indicating increased stress. • "Increased social activities": Libraries are generally quiet study spaces, not social venues. More library time would likely mean less social activities. Final answer: {"answer": ["Better time management"]}

**User**
Scenario: A company introduces remote working policies.
Condition: Employees can work from home twice a week.
Candidate factors: ["Improved work-life balance", "Reduced office costs", "More commuting"]
Task: Select the factor list that is most reasonably related to the given condition. Think step by step about each factor's relevance, then provide your selection. Please respond with your reasoning followed by Final answer: and a JSON object in this exact format: {"answer": ["factor1", "factor2", ...]}.

**Assistant**
Let me evaluate each factor: • "Improved work-life balance": Working from home twice a week allows employees to save commute time, have more flexibility, and better integrate work with personal life. This is directly related. • "Reduced office costs": With employees working from home part-time, the company needs less office space, utilities, and resources. This is a logical consequence. • "More commuting": This is contradictory--working from home twice a week would actually reduce commuting, not increase it. Final answer: {"answer": ["Improved work-life balance","Reduced office costs"]}

*Figure 11.* Few-shot prompt–response pairs for factor–condition mapping.

**Few-Shot Examples for Factor Reflection**

**System**
You are performing a self-reflection task. Given a condition and a list of initially selected factors, review each factor with a LENIENT approach. Keep factors that have ANY reasonable connection to the condition. Only remove factors that are clearly irrelevant or contradictory. When uncertain, err on the side of keeping the factor. Think step by step about each factor, then provide your reasoning followed by 'Final answer:' and a JSON array of the factors to keep.

**User**
Condition: City implements a bike-sharing program.
Initially selected factors: ["Increased bike usage", "Higher car sales", "More traffic jams"] Task: Review and keep factors reasonably related to the condition. Think step by step about each factor's relevance, then 'Final answer:' with a JSON array.

**Assistant**
"Increased bike usage" is the direct result of more shared bikes--keep it.
"Higher car sales" has no clear link to bike sharing--remove it.
"More traffic jams" could occur if road space shifts to bikes and cars interact--keep it.
Final answer: ["Increased bike usage", "More traffic jams"]

**User**
Condition: Students study in a quiet library.
Initially selected factors: ["Better concentration", "Distractions from phones", "Improved retention", "Reduced social interaction"] Task: Review and keep factors reasonably related to the condition. Think step by step about each factor's relevance, then 'Final answer:' with a JSON array.

**Assistant**
"Better concentration" follows from a quiet space aiding focus--keep it.
"Distractions from phones" contradicts the library's purpose--remove it.
"Improved retention" naturally arises from focused study--keep it.
"Reduced social interaction" is a likely side-effect of silence--keep it.
Final answer: ["Better concentration", "Improved retention", "Reduced social interaction"]

*Figure 12.* Few-shot prompt–response pairs for lenient self-reflection on factor relevance.

---

**Few-Shot Examples for Probability Estimation of Factor Support**

**User**
```
Given the scenario:  "Comparing LED bulbs vs incandescent bulbs in home lighting."
For each of the following factor values, please estimate the probability
(a float between 0 and 1) that it supports Outcome1 (LED bulbs are more advantageous)
rather than Outcome2 (incandescent bulbs are more advantageous).  Return a JSON
mapping.
Factor values:
["Initial cost per bulb", "Energy consumption per hour",
"Lifespan hours", "Color rendering index",
"Warm color temperature", "Instant full brightness",
"Dimmable compatibility", "Mercury content",
"Heat generation", "Availability"]
```

**Assistant**
```
Thought:  LEDs excel in low energy use, longevity, low heat output, and no mercury,
but cost more upfront, may have lower CRI, and vary in warmth, dimmability, and
availability.
Final answer:
{
"Initial cost per bulb": 0.30, "Energy consumption per hour": 0.95,
"Lifespan hours": 0.90, "Color rendering index": 0.25,
"Warm color temperature": 0.40, "Instant full brightness": 0.50,
"Dimmable compatibility": 0.35, "Mercury content": 0.85,
"Heat generation": 0.88, "Availability": 0.45
}
```

**User**
```
Given the scenario:  {scenario}
For each of the following factor values, please estimate the probability
(a float between 0 and 1) that it supports Outcome1:  {outcome1}
rather than Outcome2:  {outcome2}.
As reference (but not absolute) here are some initial estimates:
{prior_text}
Think step by step about each factor's relation to the outcomes,
and provide your probability estimates.  Return a JSON mapping.
Factor values:
{Factor values}
```

*Figure 13.* Few-shot prompt–response pairs for estimating the probability that a factor supports one outcome over another.

---

**Few-Shot Examples for Latent Variable Identification with CoT**

**System**
Please perform a brief chain-of-thought (step-by-step reasoning)
 before outputting the final JSON.
You are an AI assistant tasked with identifying latent variables
and assigning each latent only factors drawn from the provided list.
Do NOT output any edges. Return a JSON object with a single field:
  latents: an array of objects, each with:
    – name: string
– factors:  array of strings (each chosen from the provided Factors list)
Ensure the JSON parses correctly and strictly follows this schema.

**User**
Please identify latent variables and assign each factor to a latent.  Then return JSON
with fields:
  latents: [{"name": string, "factors": [...]}, ...]
Factors:  ["Nutrition", "Vitamins", "Taste", "Convenience"]

**Assistant**
Thought:  Nutrition and Vitamins both relate to health aspects of food, while Taste and
Convenience relate to user enjoyment and practicality.
Final answer:
{
"latents": [
  {"name": "HealthLat",   "factors": ["Nutrition","Vitamins"]},
  {"name": "EnjoyLat",    "factors": ["Taste","Convenience"]}
]
}

**User**
Please identify latent variables and assign each factor to a latent.  Then return JSON
with fields:
  latents: [{"name": string, "factors": [...]}, ...]
Factors:  ["Usability", "Security", "Maintainability", "Portability", "Reliability"]

**Assistant**
Thought:  Usability and Portability focus on user experience and access, Reliability
and Maintainability focus on software quality over time, and Security is a distinct
concern.
Final answer:
{
"latents": [
  {"name": "UXLat",      "factors": ["Usability","Portability"]},
  {"name": "QualityLat",  "factors": ["Reliability","Maintainability"]},
  {"name": "SecurityLat", "factors": ["Security"]}
]
}

*Figure 14.* Few-shot prompt–response pairs for latent variable identification with chain-of-thought reasoning.

---

**Few-Shot Examples for Latent Probability Estimation**

**System**
Please perform a brief chain-of-thought (step-by-step) before outputting the final
JSON:
You are an AI assistant. You will be given:
  • A list of latents where each latent has:
    – its name
    – the list of factor descriptions it groups
  • Two competing outcomes (Outcome1 vs. Outcome2)
Your task:  Think through the semantic content of the factors relative to the outcomes,
and estimate for each latent a probability pair [p1, p0]:
  • p1 = probability the latent supports Outcome1
  • p0 = probability the latent supports Outcome2
Begin your answer with "Thought:" to show your reasoning, then output exactly a JSON
object
mapping each latent name to its [p1, p0] (no extra text).

**User**
Latents with factors:
[
"name":"Performance", "factors":["Faster processing","Efficient resource use"],
  {"name":"Stability",   "factors":["Crash reports","Memory leaks"]}
]
Outcome1: The system improves performance.
Outcome2: The system does not improve performance.

**Assistant**
Thought:  The Performance latent groups factors that directly indicate faster and more
efficient operation, which strongly supports Outcome1.  The Stability latent lists
issues that undermine reliability, which indirectly suggests performance might not
improve overall.  I'd estimate Performance around [0.85, 0.15] and Stability around
[0.30, 0.70].
Final answer:
{
  "Performance": [0.85, 0.15],
  "Stability":   [0.30, 0.70]
}

**User**
Latents with factors:
[
  {"name":"HealthLat","factors":["Nutrition benefits"]},
  {"name":"EnjoyLat", "factors":["Taste appeal","Fun presentation"]}
]
Outcome1: Healthy eating is fun.
Outcome2: Healthy eating is not fun.

**Assistant**
Thought:  Nutrition benefits relate to health but don't guarantee fun, so HealthLat
feels neutral to slightly positive.  EnjoyLat clearly centers on taste and fun aspects,
strongly supporting Outcome1.  I'd give HealthLat [0.55, 0.45] and EnjoyLat [0.85,
0.15].
Final answer:
{
  "HealthLat": [0.55, 0.45],
  "EnjoyLat":  [0.85, 0.15]
}

*Figure 15.* Few-shot prompt–response pairs for latent probability estimation.

