# OpenReview forum: "ANCHOR: Abductive Network Construction with Hierarchical Orchestration for Reliable Probability Inference in Large Language Models"
_ICML.cc/2026/Conference — ICML 2026 regular_

### Official Review · Reviewer_zh77 · 2026-03-02

**Soundness:** 3
**Presentation:** 3
**Significance:** 2
**Originality:** 2
**Overall Recommendation:** 4
**Confidence:** 4

**Summary:**

This paper proposes ANCHOR, a multi-stage framework for improving the reliability and calibration of probability estimates produced by LLMs in contextual binary decision-making tasks. The authors argue that direct LLM probability outputs are miscalibrated and overconfident. Prior abductive–Bayesian frameworks (e.g., BIRD) generate factors via forward abduction and apply Naïve Bayes (NB) inference, but suffer from: Sparse factor spaces and Independence violations when naively expanding factors.

To address this, ANCHOR introduces three key components: 1) Bottom-up factor-space construction: Iterative LLM-based factor generation followed by UMAP + HDBSCAN clustering and LLM-guided thematic labeling, yielding a hierarchical factor space. 2) Context-aware hierarchical retrieval and filtering: Coarse-to-fine KNN retrieval over cluster prototypes and factors, followed by self-consistency voting and reflection-based filtering to reduce noise. 3) Hybrid probabilistic inference: A Naïve Bayes model with LLM-elicited likelihood proxies. A latent-augmented Causal Bayesian Network (CBN) to model factor dependencies. Aggregation via Linear Opinion Pool (LOP) or Bayesian Model Averaging (BMA).

Theoretical analysis derives closed-form CBN posteriors, justifies the proxy-likelihood assumption, and formally shows how shared latent variables induce dependence that NB cannot capture. Experiments on reasoning/planning datasets (Common2Sense, Today, Plasma) and fact-checking benchmarks (COVID, ExpertQA, CNN, XSum) demonstrate: 1) Improved F1 in preference-based pairwise evaluation. 2) Consistent accuracy gains over baselines in decision-making. 3) Near-complete coverage with reduced “unknown” predictions. 4) Lower token/time cost compared to BIRD.

This research strives to study the concept of structured probabilistic reasoning layered on top of LLM-generated explanatory abstractions, aiming to transform opaque confidence signals into calibrated, auditable Bayesian inference pipelines. Overall, this article's core idea pertains to integrating hierarchical knowledge construction with explicit Bayesian graphical modeling to improve the reliability and interpretability of LLM-based probability estimation.

**Compliance With Llm Reviewing Policy:**

Affirmed.

**Final Justification:**

I have adjusted the score accordingly.

**Key Questions For Authors:**

- How stable are the discovered latent structures across runs and prompts? Have you measured structural consistency metrics?

- Can you evaluate calibration on datasets with global label comparability and repeated sampling?

- Do human evaluators find the factor hierarchy more interpretable than BIRD’s factor space?

- Can ANCHOR extend beyond binary inference without exponential complexity growth?

**Limitations:**

- The hierarchical factor space is scenario-specific and may not scale to large, open-domain knowledge bases.

- Latent modeling introduces additional epistemic uncertainty not quantified.

- KNN retrieval may fail under embedding drift.

- CBN exact inference may become expensive as latent groups grow.

- No comparison with neural calibration methods (e.g., temperature scaling, conformal prediction).

**Strengths And Weaknesses:**

Strengths:
1) The paper clearly identifies two central weaknesses in prior abductive frameworks: Factor sparsity leading to abstention. Factor over-expansion breaking conditional independence.

2) The three-stage pipeline is well-structured: Generation → Organization → Retrieval → Inference. Unlike many LLM-based systems that rely purely on prompting heuristics, ANCHOR builds a persistent structured factor space. This design is principled and reusable across contexts.

Weaknesses:
1) The method depends critically on: Factor-level posterior elicitation,  Latent conditional probabilities Latent discovery itself. The system’s reliability is fundamentally tied to LLM probability quality. The calibration study suggests mild sensitivity, but deeper analysis of failure modes is missing.

2) Latent variable grouping is entirely LLM-driven. There is no evaluation of: Structural accuracy,  Stability across prompts, Inter-run variance of latent structure. The causal interpretation of the CBN remains heuristic rather than empirically validated.

3)The dataset consists of small pairwise subgraphs. ECE-style calibration metrics are unsuitable. Therefore, strong claims about “calibration” may be overstated. The evaluation focuses more on ranking consistency than true probabilistic calibration.

---

> ### Author Rebuttal · Authors · 2026-03-27
>
> We thank the reviewer for the constructive feedback and for recognizing the clear motivation, structured design, and technical soundness of our work. Due to space limits, we keep our response brief and are happy to clarify further if needed.
>
> >  W1
>
> We agree that ANCHOR depends on factor-level posteriors, latent probabilities, and latent discovery, but it is not inherently tied to LLM-derived priors: as shown in Table 4 and Appendix B.7, it can also learn probabilistic parameters from data and still achieve strong performance. To better understand failures, we defined five error categories and used Qwen3-80B-A3B to assign a primary cause to each misclassified case. The main errors come from factor grounding error (wrong factor meaning) and overconfident probability bias (overly confident prob), while the rest are due to insufficient factor separation (two sides too similar), weak factor-set preference (no clear support), and missing key latent factors (important factors missing).
>
> |Failure mode|rate|
> |---|---:|
> |Insufficient factor separation|10.1%|
> |Overconfident probability bias|28.7%|
> |Weak factor-set preference|18.2%|
> |Missing key latent factors|8.7%|
> |Factor grounding error|34.3%|
>
>
> > W2
>
> We test ANCHOR under three prompt variants: compact rephrasing, schema-first reordering, and format-only reminder, while keeping the same latent-grouping task.We report cross-prompt ARI and NMI against the original prompt to measure grouping consistency. ANCHOR remains fairly stable under prompt changes, with a larger drop only under stricter formatting constraints.
>
> |Variant|ARI|NMI|
> |---|---:|---:|
> |Original|1.000|1.000|
> |Compact rephrasing|0.690|0.772|
> |Schema-first reordering|0.680|0.772|
> |Format-only reminder|0.572|0.666|
>
> We test structure sensitivity by perturbing the learned CBN and find mostly moderate degradation, suggesting that ANCHOR is reasonably robust to structural changes, with the larger drop on PLASMA indicating greater benefit when factor interactions are more structured.
>
> |Variant|CNN|EXPERTQA|PLASMA|TODAY|
> |---|---:|---:|---:|---:|
> |ANCHOR|62.9|60.5|76.2|82.7|
> |Drop-One-Latent|60.2|58.9|74.1|81.3|
> |Merged-Latents|58.4|56.7|67.4|77.2|
>
>
> > w3
>
> We evaluate cross-dataset calibration in fact-checking with Qwen2.5-72B-Ins, fitting post-hoc calibrators on covid and xsum and testing on cnn and expertqa. Using next-token logits for binary labels (support/unsupport), we compare temperature scaling, isotonic regression, and histogram binning against ANCHOR-Qwen2.5-72B. ANCHOR performs best on both target datasets across all three metrics.
>
> |Data|Method|BalancedAcc|Brier|ECE|
> |---|---|---:|---:|---:|
> |CNN|TS|0.500|0.3311|0.2985|
> |-|IR|0.613|0.4467|0.1530|
> |-|HB|0.556|0.4879|0.4170|
> |-|Anchor|**0.621**|**0.2458**|**0.1472**|
> |ExpertQA|TS|0.523|0.4175|0.1444|
> |-|IR|0.490|0.6248|0.2552|
> |-|HB|0.514|0.5405|0.2940|
> |-|Anchor|**0.611**|**0.3702**|**0.1021**|
>
>
> >Q1
>
> Same as W2
>
> > Q2
>
> Same as W3
>
> > Q3
>
> We conducted a small human preference study with 30 participants over 10 representative questions, where they compared ANCHOR’s hierarchical factor space with BIRD’s flatter factor space without knowing which method each option came from. ANCHOR was preferred on all 10 questions, with an overall preference rate of 61.3% versus 34.0% for BIRD and 4.7% ties, providing preliminary evidence that ANCHOR’s factor hierarchy is more interpretable and explanatory
>
> > Q4
>
>
> Yes. ANCHOR can extend beyond binary inference without exponential class growth by using a one-vs-rest design: the factor space and mapping pipeline are reused, and only the final decision layer is expanded to one binary head per class, so cost grows roughly linearly with the number of classes.
>
>
> >  L1
>
> ANCHOR is currently designed for scenario-level contextual inference, where a focused hierarchical factor space is especially effective, while extension to open-domain settings with retrieval-based factor reuse and scalable indexing remains a promising future direction.
>
> >  L2
>
> Latent discovery and latent probability elicitation naturally introduce an additional source of uncertainty. While our current stability and sensitivity analyses offer some preliminary evidence of robustness, a more explicit treatment of uncertainty in the latent layer is an important direction for future work
>
> >  L3
>
> This is a valid but general concern for retrieval-based methods, and in ANCHOR its impact appears limited since KNN retrieval is only an initial step and later filtering further reduces retrieval noise.
>
> > L4
>
> We believe this issue is modest in our setting. ANCHOR performs inference only on the compact mapped factor set $F^*(u)$, with a simple $Outcome \rightarrow L_i \rightarrow f_j$ structure; we also show that the likelihood decomposes by latent groups and the posterior has a closed form. In practice, the main risk is uncontrolled graph expansion, which can be controlled by limiting latent groups, capping group size, or pruning weak factors.
>
> > L5
>
> Please refer to W3

---

> > ### Author Rebuttal · Reviewer_zh77 · 2026-04-01
> >
> > The rebuttal responds to my concerns.

---

> > > ### Author Response · Authors · 2026-04-01
> > >
> > > Thank you for the update. We noticed that you marked the acknowledgement as fully resolved, and we are very glad that our rebuttal was helpful in addressing your concerns. We sincerely appreciate your time and thoughtful feedback.

---

### Official Review · Reviewer_qNfW · 2026-03-04

**Soundness:** 3
**Presentation:** 2
**Significance:** 3
**Originality:** 3
**Overall Recommendation:** 4
**Confidence:** 4

**Summary:**

This paper is about decision making under incomplete information where a large number of factors play a role. The belongs to the school of thought that attempts to develop an approach based on LLM-based information and data gathering combined with Bayesian Network analysis. The paper's contribution is an improvement in determining the factor space and the integration of a causal BN with latent variables to accommodate dependencies between variables. A comparison with existing approaches shows some improvement and an ablation study shows that all components in the ANCHOR approach have added value.

**Compliance With Llm Reviewing Policy:**

Affirmed.

**Final Justification:**

The clarifications in the rebuttal and in the new version of the paper improved my understanding of the approach considerably. As a consequence, I have updated my presentation and originality scores, as well as the overall recommendation to "weak accept".

**Key Questions For Authors:**

Q1: Figure 2 has 6 LLM-boxes while there are 9 prompts in Appendix C.3. How do these 9 map to the 6 boxes?
If only you would simply have provided a reference to the appendix there to this appendix, I would have gained conceptual understanding of the approach much earlier. There are actually barely (no?) references to the appendices, so the reader doesn't know what additional material is there and where the additional material belongs to.
Q2: You report Ctx.1 and Ctx.2 scores 0.610 and 0611, Same score 0.278, Avg score 0.606 and Cov almost 99.95% (Table 1, bottom row). These are not perfect, of course, but do they suffice for application in practice? If not, how far are we off? Is 0.63 completely unusable or fine with adequate carefulness and after-checks? What are the implications of ANCHOR making different kinds of mistakes?
Q3: Table 3 reports Ctx.1 and Ctx.2 scores 0.626 and 0.627, Same score 0.281, Avg score 0.596 and Cov 99.95% for ANCHOR (full) and Qwen2.5-72b. These are not the same as in Table 1. Why not? Seems like the same conditions. Or is this just a re-run of the same experiment that resulted in slightly different results?
Q4: How do you arrive at $U_1$ and $U_2$ in Section 5.1 paragraph on Preference-based Pairwise Evaluation? This is probably described in the benchmark paper, but for the paper to stand on its own, you may want to say a few more words on this (and perhaps provide or refer to an example) to provide the reader with some idea on how challenging the task in the benchmark is. For all we know, it is a non-representative easy task for which good results mean nothing (don't believe so, but the paper doesn't provide convincing argumentation).
Q5: In 4.3.1, you describe that you obtain probabilities from the LLM. You refer to certain papers evaluating this, but they probably found out that it is not perfect ... are these probability estimates sufficiently good for your purposes? What quality do you actually need?

**Limitations:**

No discussion of limitations.
What I would have liked discussed are limitations in the experimental design as well as the implications of imperfect results (see Q2).

**Strengths And Weaknesses:**

Strong points:
Large-scale decision making under incomplete information is a very relevant problem to work on. Employing LLMs for this problem is obviously an attractive endeavor. The combination with Bayesian Network modeling and analysis is the important and significant novelty (not only from this paper alone but your research line).
The technical soundness is fine: exact descriptions with effective notation, experiments well-designed (I especially like the ablation study: provides good added value).

Weak points:
Main weak point is the presentation. From the main paper, I had major trouble understanding what this was all about. The bigger picture and motivation are completely missing. The paper immediately dives deep (abstract does so as well). I had to fully struggle through the 26 pages (!!!) of appendices to understand the bigger picture for which this paper is but a step. 12 pages paper with 26 pages appendix is also rather ill-balanced. One thing that I still have trouble with understanding is the role of the "downstream condition" in the framing of the problem.
Figure 1 could be seen as an attempt, but the figure has no text that uses the figure to explain the conceptual idea and bigger picture. A more conceptual explanation is direly need as well as an example such that the reader knows what you are talking about with $C$, $O_1$, $O_2$, $u$, $S_{cen}$, and $\mathcal{F}$.

---

> ### Author Rebuttal · Authors · 2026-03-27
>
> We thank the reviewer for the constructive feedback and for recognizing the importance, novelty, and technical soundness of our work. Due to space limits, we keep our response brief and are happy to clarify further if needed.
>
> > # W1
>
> We apologize for the appendix burden. Most appendix content is supplementary and intended for reproducibility rather than core understanding. We agree the linkage should be clearer, and in the revision we will add explicit pointers from the main text to the relevant appendix parts, e.g., Sec. 3->App. A.1/C.1, Sec. 4.3 -> App. A.2--A.4, Sec. 5.1 -> App. B.3/B.4, Sec. 5.3 -> App. B.1/B.2, Sec. 5.4-> App. B.5--B.7, and Fig. 2 -> App. C.3.
> > # W2
>
> For example, consider the scenario $S_{cen}$ of comparing whether noodles cook more efficiently in hot water or warm water. The two candidate outcomes are $O_1$: noodles cook more efficiently in hot water, and $O_2$: noodles cook more efficiently in warm water. A downstream condition $u$ provides partial evidence for this scenario. For instance, $u_1$ may be “the higher temperature of hot water accelerates starch gelatinization,” while $u_2$ may be “hot water improves water absorption.” These define two contexts, $C_1=(S_{cen},u_1)$ and $C_2=(S_{cen},u_2)$.
>
> Given $S_{cen}$, the model first constructs a scenario-level factor space $\mathcal{F}$, such as temperature, hydration, starch breakdown, and cooking efficiency. It then maps each condition $u$ to the relevant factors in $\mathcal{F}$ and infers how strongly these factors support $O_1$ versus $O_2$. In this example, both $u_1$ and $u_2$ support $O_1$, but possibly with different strengths. If $u_1$ is stronger than $u_2$, the desired result is that $P(O_1\mid C_1) > P(O_1\mid C_2)$, while both remain higher than the corresponding probabilities for $O_2$.
>
> We will add this to Section 1 (Introduction) as a concrete illustration to further explain Figure 1 and the roles of $S_{cen}$, $u$, $C$, $O_1$, $O_2$, and $\mathcal{F}$.
>
> > # Q1
>
> Specifically, the prompts in Appendix C.3 correspond to the modules in Figure 2 as follows: Figures 7-8 for Iterative Factor Generation, Figure 9 for Majority Vote, Figure 10 for Thematizing Clusters, Figure 11 for condition-to-factor mapping, Figure 12 for Self Reflection, Figure 13 for Prior Probability Prediction, Figure 14 for Latent Variable Identification, and Figure 15 for Latent Probability Evaluation. Appendix as W1.
>
> > # Q2
>
> We believe the results are already meaningful for decision support, though not yet for fully autonomous high-stakes use. High coverage does not guarantee high performance, since the core challenge is still fine-grained probabilistic inference rather than factor retrieval alone; this is why we compare against fully covered **factor-based methods**. The task is also genuinely difficult—ANCHOR outperforms 671B-scale open-weight models, while even o1 struggles in BIRD—and we use NB + CBN aggregation to reduce the risk of confident but incorrect estimates.
>
> > # Q3
>
> Thank you for pointing this out. Table 1 reports the mean and standard deviation over five runs, while the ablation results in Table 3 are based on one representative run from the same experimental setting. Therefore, the ANCHOR (full) row in Table 3 is not expected to exactly match the averaged results in Table 1. We will make this explicit in the table caption to avoid confusion.
>
> > # Q4
>
> Thank you for this question. In the BIRD/Common2Sense pairwise setup, each test instance is already given as $(S_{cen}, O_1, O_2, U_1, U_2)$: a scenario, two candidate outcomes, and two pre-constructed conditions from the dataset. For example, in a noodle-cooking scenario, $O_1$ may be “hot water cooks noodles more efficiently,” $O_2$ “warm water cooks noodles more efficiently,” while $U_1$ is “higher temperature accelerates starch gelatinization” and $U_2$ is “hot water improves water absorption.”
>
> Thus, $U_1$ and $U_2$ are not created by our method; we directly use the pairwise data provided by BIRD. Importantly, the model does not compare $U_1$ and $U_2$ directly. Instead, it evaluates them independently as $C_1=S_{cen}+U_1$ and $C_2=S_{cen}+U_2$, predicts $P(O_i\mid C_1)$ and $P(O_i\mid C_2)$ separately, and then compares these probabilities to determine which condition provides stronger support. This is what makes the task challenging: both conditions may support the same outcome, so the model must recover fine-grained differences in support strength rather than simply predict the correct label.
>
> > # Q5
>
> ANCHOR does not require perfectly calibrated LLM probabilities; it only needs a coarse but discriminative signal to initialize NB + CBN inference. Empirically, Appendix B.5.2 shows limited sensitivity to the estimator LLM: replacing Qwen2.5-72B with GPT-4o-mini changes results by only about $\pm 2.5$ points. We also visualize the factor-level probability distributions, the overall trends remain broadly consistent, which helps explain why the results are not strongly affected.

---

> > ### Author Rebuttal · Reviewer_qNfW · 2026-04-05
> >
> > The clarifications of the authors provided a few crucial points in fully grasping the method. I will update my scores accordingly.

---

> > > ### Author Response · Authors · 2026-04-05
> > >
> > > We really appreciate your careful reading and thoughtful feedback, and we’re glad our rebuttal helped clarify the paper. Thank you again for taking the time to revisit the work and update your assessment.

---

### Official Review · Reviewer_xAvb · 2026-03-12

**Soundness:** 3
**Presentation:** 3
**Significance:** 3
**Originality:** 3
**Overall Recommendation:** 4
**Confidence:** 3

**Summary:**

The paper studies generating reliable probability estimates from LLMs. The paper proposes ANCHOR, a framework that constructs a dense, hierarchical factor space and employs Causal Bayesian Networks to improve calibration and reduce "unknown" outcomes, achieving state-of-the-art performance with high efficiency.

**Compliance With Llm Reviewing Policy:**

Affirmed.

**Key Questions For Authors:**

What is the impact of each hyperparameter (e.g., K1, K2, and alpha)? These may require recalibration when applying ANCHOR to different data distributions.

**Limitations:**

yes

**Strengths And Weaknesses:**

Strengths:
1. ANCHOR significantly reduces the unknown outcome rate compared to baselines. It ensures that even nuanced or specific conditions find relevant explanatory factors by proactively building a dense factor library and using hierarchical retrieval.
2. The model relaxes the rigid (and often violated) independence assumption of Naïve Bayes, which is theoretically sound.
3. Despite the complexity of the framework, the authors demonstrate that it reduces downstream inference time and token usage (approx. 0.24× the tokens of BIRD).
4. The paper evaluates the framework across diverse tasks, including reasoning, planning, and fact-checking (Common2Sense, Plasma, Today, XSum, etc.). The results show state-of-the-art performance and a superior alignment with human preferences.

Weaknesses
1. System Complexity: The ANCHOR pipeline is multi-staged, involving iterative generation, HDBSCAN clustering, UMAP projection, KNN retrieval, and dual-model (NB+CBN) aggregation. This complexity might make it challenging to implement and debug in production environments compared to simpler prompting strategies.
2. Reliance on LLM Priors: While the framework structures the reasoning process, it still relies on the LLM to elicit conditional probability tables. If the underlying LLM has systemic biases or extreme miscalibration regarding certain facts, these errors could still propagate through the Bayesian network.
3. Cold-Start Overhead: While the per-scenario cost is lower, the initial iterative factor-space construction stage represents a significant computational and token cost. This makes the system better suited for persistent domains rather than one-off, highly unique queries.

---

> ### Author Rebuttal · Authors · 2026-03-27
>
> We thank the reviewer for the careful reading and helpful feedback. We appreciate the recognition of ANCHOR’s reduced unknown rate, its principled relaxation of the Naïve Bayes independence assumption, its efficiency gains, and its strong results across diverse tasks. Due to space limits, we briefly address the main concerns and are happy to clarify further if needed.
>
>
>
> >  w1 System Complexity
>
> We appreciate this concern. Although ANCHOR has multiple stages, its modules are functionally decoupled: factor generation, factor organization, context–factor mapping, and probabilistic inference can each be inspected and tested separately, which makes the system easier to analyze and debug than an end-to-end black box. We also examined robustness empirically. As shown in Appendix B.5.1, clustering quality can be assessed using metrics such as Silhouette Score, and different clustering methods can be chosen for different scenarios rather than relying on a single fixed algorithm. Appendix B.6 further shows low sensitivity to the retrieval backend. Together with the ablations in Table 3, these results suggest that ANCHOR is structured but not brittle: modules can be tested or replaced independently, and changes lead to gradual rather than catastrophic degradation.
>
> > w2 Reliance on LLM Priors
>
> We thank the reviewer for this important point, which we explicitly considered in our design. Some dependence on LLM-derived priors is difficult to avoid in this line of work: even BIRD learns its CPTs from LLM-labeled data, so this issue is not unique to ANCHOR. Our design reduces this dependence by using the LLM only for coarse, interpretable support signals for factor- and latent-level parameterization, rather than requiring finely calibrated probability tables from a single model. We also evaluate this empirically: Appendix B.5.2 shows that replacing the probability-estimator LLM from Qwen2.5-72B to GPT-4o-mini, while keeping the factor space, mapping, and NB+CBN inference fixed, causes only modest performance changes across CNN, Today, Plasma, and ExpertQA. Importantly, ANCHOR is also not inherently tied to LLM priors: as shown in Table 4 and Appendix B.7, it can also learn probabilistic parameters from data, in the same spirit as BIRD-style CPT training, and still achieve strong performance. Thus, while LLM bias may still propagate to some extent, this is a general challenge rather than a specific weakness of ANCHOR, and our framework already supports a practical data-driven alternative.
>
> > w3 Cold-Start Overhead
>
> We thank the reviewer for this important point. At the current research stage, ANCHOR is designed around scenario-specific factor construction, so an upfront cost is indeed incurred when building the factor space for a new scenario. However, this is also where ANCHOR’s main advantage lies: once factors are constructed, the hierarchical organization and mapping mechanism make them easier to reuse, extend, and maintain than a flat factor pool. This makes the framework naturally transferable to more general domains through structured reuse, hierarchical organization, and incremental updates of the factor library, rather than rebuilding everything from scratch for each new query. In this sense, the cold-start cost is a practical trade-off of the current design, but it also reflects one of ANCHOR’s strengths: the factor space can evolve into a reusable knowledge structure that reduces future overhead. We will clarify this point more explicitly in the revision.
>
> > Q1 What is the impact of each hyperparameter (e.g., K1, K2, and alpha)? These may require recalibration when applying ANCHOR to different data distributions.
>
> We thank the reviewer for this important question. We analyze the sensitivity of $K_1$ and $K_2$ in Appendix B.6, where we study how behavior changes with different $K$ values. At the same time, ANCHOR is not tied to a single retrieval backend: in different scenarios, it can naturally switch to other retrieval methods (e.g., BM25 or FAISS) that do not require the same manual hyperparameter choices, which helps reduce recalibration burden under distribution shift.
>
> For $\alpha$, its role is interpretable: it controls the trade-off between coarse-grained theme-level signals and fine-grained factor-level signals. In practice, this can be selected on a validation set for a given scenario. More generally, such combination weights can also be learned from held-out data rather than fixed manually. Concretely, on a held-out dev set, we fit a logistic regression that takes the NB margin and CBN margin as two features. The learned coefficients are positive and non-zero for both backbones, indicating that NB and CBN provide complementary signals and that learning the combination is a reasonable design. Thus, while some hyperparameters may need adaptation across data distributions, ANCHOR provides both sensitivity analysis and simple validation-based ways to recalibrate them.

---

> > ### Author Rebuttal · Reviewer_xAvb · 2026-03-31
> >
> > I thank the authors for their comprehensive rebuttal. My concerns regarding system complexity, LLM priors, and hyperparameters have been mostly addressed, and I will keep my scroe.

---

> > > ### Author Response · Authors · 2026-04-01
> > >
> > > We sincerely thank the reviewer for the thoughtful follow-up and for recognizing that our rebuttal has addressed the main concerns. We greatly appreciate your careful consideration. Regardless, we very much appreciate your time and helpful feedback.

---

### Official Review · Reviewer_c3SC · 2026-03-14

**Soundness:** 3
**Presentation:** 3
**Significance:** 3
**Originality:** 2
**Overall Recommendation:** 4
**Confidence:** 4

**Summary:**

This paper introduces ANCHOR, an inference framework for aggregated Bayesian reasoning over a hierarchically structured factor space constructed through bottom-up abduction, clustering, and LLM-guided theming. It then conducts a context-aware mapping using hierarchical retrieval and uses both a Nave Bayes model and a Causal Bayesian Network to capture latent dependencies between factors.  Experiments show that ANCHOR eliminates “unknown” predictions, improves both coverage and predictive quality over prior Bayesian-LLM baselines, and maintains  inference efficiency.

**Compliance With Llm Reviewing Policy:**

Affirmed.

**Key Questions For Authors:**

How much of the improvement comes from better factor generation versus better mapping versus better inference?

Why is the hierarchical factor-space construction methodically necessary? To what extent do the gains come from the hierarchical organization itself, as opposed to simply constructing a larger and denser factor pool with better coverage?

**Limitations:**

See weaknesses.

**Strengths And Weaknesses:**

Strengths:

1. The paper tackles an important problem: reducing abstentions and unreliable outputs in probabilistic reasoning settings where LLMs are used together with external structured inference components.

2. The experiments are extensive with detailed ablation studies.

Weaknesses:

1. I have novelty concerns as this paper is primarily an extension and refinement of BIRD rather than a more substantial conceptual advance.

2. As described in Section 4.3, the framework still relies on LLM-elicited probabilities for factor- and latent-level parameterization. This may be practical, but it weakens the claim that the external Bayesian layer is the main source of reliability, since the final inference still depends on LLM probability judgments. The authors should better justify this.

3. There are many moving pieces in the proposed pipeline, the authors should justify the design choices.  1) How much of the improvement comes from better factor generation versus better mapping versus better inference?  2) Why is the hierarchical factor-space construction methodically necessary? To what extent do the gains come from the hierarchical organization itself, as opposed to simply constructing a larger and denser factor pool with better coverage?

---

> ### Author Rebuttal · Authors · 2026-03-27
>
> We thank the reviewer for the careful reading and helpful feedback. We appreciate the recognition of the importance of this problem and of our extensive experiments and ablations. Below, we briefly address the main concerns and are happy to clarify further if needed
>
> > W1
>
> We appreciate the reviewer’s concern. We would like to first emphasize that BIRD is a strong and important foundation for this research line: it established the core paradigm of combining LLM-generated abductive factors with Bayesian inference for more trustworthy probability estimation, and showed that this direction can improve reliability over direct LLM confidence.  Building on this foundation, our contribution is not only an empirical refinement but a methodological innovation with broader future implications. ANCHOR introduces a hierarchical and reusable factor-space construction, a structured retrieval-based mapping mechanism, and a dual NB+CBN inference layer for modeling latent factor dependencies; together, these make the framework more scalable, more robust to sparse factor spaces, and better suited for calibrated decision support. Because ANCHOR builds a persistent and reusable reasoning structure rather than solving each case only reactively, we believe its impact extends beyond BIRD-style refinement: it can serve as a stronger base design for future larger-scale, higher-stakes LLM-Bayesian systems where interpretability, auditability, and calibrated probabilities are all essential
>
> > W2
>
> We thank the reviewer for this important point, which was explicitly considered in our design. Some dependence on LLM-derived priors is difficult to avoid in this line of work: even BIRD learns its CPTs from LLM-labeled data, so this issue is not unique to ANCHOR. Our goal is not to ask the LLM for finely calibrated probabilities, but only for coarse, interpretable support signals for factor- and latent-level parameterization, thereby reducing reliance on fragile probability judgments from any single backbone. We also evaluate this empirically: Appendix B.5.2 shows that replacing the probability-estimator LLM from Qwen2.5-72B to GPT-4o-mini, while keeping the factor space, mapping, and NB+CBN inference fixed, causes only modest performance changes across CNN, Today, Plasma, and ExpertQA. ANCHOR’s robustness does not rely on finely calibrated probabilities from a specific LLM, but on how coarse signals are structured and propagated through the Bayesian models. As discussed in Appendix B.7, LLM-based priors are a practical initialization when full supervision is unavailable, though they may still inherit LLM bias. We will clarify this in the revision.
>
> > W3
>
> We thank the reviewer for this important question. Our results suggest that the gains do not come from a single component. In Sec. 5.3, replacing BIRD’s factor space with ANCHOR’s factors improves performance (0.513→0.532), adding ANCHOR’s mapping preserves the gain at much lower cost (0.526), and adding CBN aggregation gives the best result (0.568), showing separate contributions from factor construction, mapping, and inference. Our ablations also show that the gains are not simply from a larger factor pool: w/o cluster, which directly expands factors without hierarchical clustering, causes the largest drop (0.596→0.400 on Qwen2.5-72B), while w/o hierarchy, which keeps clustering but removes weighted fusion and coarse-to-fine retrieval, also drops clearly (0.596→0.491). We also include a Factor-based setting that uses gold-outcome-aware factors to simulate an idealized large-factor-space case, yet its performance is still limited. Together, these results suggest that ANCHOR’s gains come from structured organization, mapping, and inference, rather than simply more factors
>
> > Q1
>
> We believe the gains come from all three components, not any single one: factor generation improves evidence quality and coverage, mapping selects relevant factors more efficiently, and inference improves final decisions by modeling factor dependencies. We provide detailed evidence in our response to W3
>
> > Q2
>
> We believe the hierarchical factor space is necessary, not just a larger factor pool. Prior work shows that coarse-to-fine and hierarchical retrieval improve both quality and efficiency, and that modeling dependencies matters when evidence is correlated [1][2][3][4]. Consistent with this, as discussed in W3, our gains do not come simply from more factors or better coverage, but from hierarchical organization, better mapping, and dependency-aware inference: directly enlarging the factor set is insufficient, and removing hierarchical retrieval/fusion also clearly hurts performance.
>
> [1] Coarse-to-Fine Question Answering for Long Documents (Choi et al., ACL 2017)
>
> [2] Sarthi et al., RAPTOR: Recursive Abstractive Processing for Tree-Organized Retrieval 2024.
>
> [3] Retrieval-Augmented Generation with Hierarchical Knowledge (Huang et al., Findings 2025)
>
> [4] Friedman et al., Bayesian Network Classifiers, 1997.

---

> > ### Author Rebuttal · Reviewer_c3SC · 2026-04-03
> >
> > I am satisfied with the authors' rebuttal and I will keep my score.

---

> > > ### Author Response · Authors · 2026-04-03
> > >
> > > We sincerely appreciate your careful reading of our paper and your valuable comments. We are glad that our rebuttal helped address your concerns.

---

### Decision · Program_Chairs · 2026-04-30

**Decision:**

Accept (regular)

**Comment:**

This paper proposes a framework for reliable probability inference that uses LLMs to generate structured reasoning factors and performs inference via Bayesian networks. All reviewers remain positive about this submission. Prior to the rebuttal, reviewers agreed that the paper tackles an important problem (c3SC, qNfW, zh77), the proposed method is effective (xAvb), and the experiments and ablations are well designed (qNfW, xAvb, c3SC). Meanwhile, reviewers raised common concerns about the reliance on LLM-derived probabilities (c3SC, xAvb, zh77), insufficient justification for several design choices (c3SC, xAvb, qNfW, zh77), and presentation clarity (qNfW). During the rebuttal, the authors provided point-by-point clarifications, and after rebuttal, all reviewers indicated that their concerns had been resolved. Given the consensus among the reviewers, the AC recommends weak accept.